# DUCT reveals architectural mechanisms contributing to bile duct recovery in a mouse model for Alagille syndrome

**Simona Hankeova[1,2†], Jakub Salplachta[3†], Tomas Zikmund[3‡], Michaela Kavkova[3‡], Noémi Van Hul[1‡], Adam Brinek[3], Veronika Smekalova[3], Jakub Laznovsky[3], Feven Dawit[4], Josef Jaros[5], Vítězslav Bryja[2], Urban Lendahl[6], Ewa Ellis[4], Antal Nemeth[7], Björn Fischler[4], Edouard Hannezo[8], Jozef Kaiser[3*], Emma Rachel Andersson[1,6*]**

[1]Department of Biosciences and Nutrition, Karolinska Institutet, Solna, Sweden; [2]Department of Experimental Biology, Masaryk University, Brno, Czech Republic; [3]CEITEC – Central European Institute of Technology, Brno University of Technology, Brno, Czech Republic; [4]Department of Pediatrics, Clinical Science, Intervention and Technology (CLINTEC), Karolinska Institutet and Karolinska University Hospital, Solna, Sweden; [5]Department of Histology and Embryology, Masaryk University, Brno, Czech Republic; [6]Department of Cell and Molecular Biology, Karolinska Institutet, Solna, Sweden; [7]Department of Laboratory Medicine, Karolinska Institutet, Solna, Sweden; [8]Institute of Science and Technology, Klosterneuburg, Austria

**\*For correspondence:**
Jozef.Kaiser@ceitec.vutbr.cz (JK);
emma.andersson@ki.se (ERA)

[†]These authors contributed equally to this work
[‡]These authors also contributed equally to this work

**Competing interests:** The authors declare that no competing interests exist.

**Abstract** Organ function depends on tissues adopting the correct architecture. However, insights into organ architecture are currently hampered by an absence of standardized quantitative 3D analysis. We aimed to develop a robust technology to visualize, digitalize, and segment the architecture of two tubular systems in 3D: *double resin casting micro computed tomography* (DUCT). As proof of principle, we applied DUCT to a mouse model for Alagille syndrome (*Jag1[Ndr/Ndr]* mice), characterized by intrahepatic bile duct paucity, that can spontaneously generate a biliary system in adulthood. DUCT identified increased central biliary branching and peripheral bile duct tortuosity as two compensatory processes occurring in distinct regions of *Jag1[Ndr/Ndr]* liver, leading to full reconstitution of wild-type biliary volume and phenotypic recovery. DUCT is thus a powerful new technology for 3D analysis, which can reveal novel phenotypes and provide a standardized method of defining liver architecture in mouse models.

## Introduction

The correct three-dimensional (3D) architecture of lumenized structures in our bodies is essential for function and health. The cardiovascular system, lungs, kidneys, liver, and other organs depend on precisely patterned tubular networks. Several diseases are caused by, or result in, alterations in the 3D architecture of lumenized structures. Vascular architecture defects contribute to Alzheimer's disease (*Klohs et al., 2014*), opportunistic infections cause narrowing of bile ducts in liver (*De Angelis et al., 2009*), and branching morphogenesis defects in the renal urinary system cause hypertension (*Short and Smyth, 2016*). In some pathologies, several lumenized structures are affected at once. Visualizing multiple tubular systems in tandem in 3D, in animal disease models, is necessary to allow investigation of how these systems interact in vivo in development, homeostasis, and disease.

**eLife digest** Many essential parts of the body contain tubes: the liver for example, contains bile ducts and blood vessels. These tubes develop right next to each other, like entwined trees. To do their jobs, these ducts must communicate and collaborate, but they do not always grow properly. For example, babies with Alagille syndrome are born with few or no bile ducts, resulting in serious liver disease. Understanding the architecture of the tubes in their livers could explain why some children with this syndrome improve with time, but many others need a liver transplant.

Visualising biological tubes in three dimensions is challenging. One major roadblock is the difficulty in seeing several tubular structures at once. Traditional microscopic imaging of anatomy is in two dimensions, using slices of tissue. This approach shows the cross-sections of tubes, but not how the ducts connect and interact. An alternative is to use micro computed tomography scans, which use X-rays to examine structures in three dimensions. The challenge with this approach is that soft tissues, which tubes in the body are made of, do not show up well on X-ray. One way to solve this is to fill the ducts with X-ray absorbing resins, making a cast of the entire tree structure. The question is, can two closely connected tree structures be distinguished if they are cast at the same time?

To address this question, Hankeova, Salplachta et al. developed a technique called double resin casting micro computed tomography, or DUCT for short. The approach involved making casts of tube systems using two types of resin that show up differently under X-rays. The new technique was tested on a mouse model of Alagille syndrome. One resin was injected into the bile ducts, and another into the blood vessels. This allowed Hankeova, Salplachta et al. to reconstruction both trees digitally, revealing their length, volume, branching, and interactions. In healthy mice, the bile ducts were straight with uniform branches, but in mice with Alagille syndrome ducts were wiggly, and had extra branches in the centre of the liver.

This new imaging technique could improve the understanding of tube systems in animal models of diseases, both in the liver and in other organs with tubes, such as the lungs or the kidneys. Hankeova, Salplachta et al. also lay a foundation for a deeper understanding of bile duct recovery in Alagille syndrome. In the future, DUCT could help researchers to see how mouse bile ducts change in response to experimental therapies.

2D histological sections remain the standard practice for all types of tissues. Recent advances in tissue clearing (*Chung et al., 2013*; *Susaki et al., 2014*; *Renier et al., 2016*) carbon ink injections (*Kaneko et al., 2015*) and high-end microscopy begin to address the need for, and benefits of, whole organ analysis. Importantly, organ systems often interact with one another and almost always are connected to blood supply. In order to study tissue spatial organization in development, disease, and regeneration, a 3D analysis of multiple networks is indispensable. A lack of markers, suitable antibodies, tissue autofluorescence and/or organ size often preclude the possibility for whole organ analysis. Radiopaque resin casting is an alternative approach that enables micro computed tomography (μCT) scanning, digitalization with full rotation and the possibility for both qualitative and quantitative analyses compatible with multiple imaging softwares.

In this study, we focused on establishing a simple, robust, antibody-free and inexpensive method for whole organ visualization of lumenized structures. We built on previous work to image a single network using resin (*Masyuk et al., 2003*; *Kline et al., 2011*; *Walter et al., 2012*) and imaging with μCT. First, in order to visualize multiple structures, we tested different radiopaque substances to enhance the resin contrast, resulting in mixing of two MICROFIL resins with distinctive radiopacity. As proof of principle, we focused on studying the hepatic vascular and biliary systems of wild type and *Jagged1* Nodder (*Jag1^{Ndr/Ndr}*) (*Andersson et al., 2018*) mice, which are challenging to image by other methods due to liver size and autofluorescence (*Renier et al., 2016*). We devised *double resin-casting micro computed tomography* (DUCT) to inject, image and digitalize two systems in tandem in 3D to gain a deeper insight into the organ recovery process. The subsequent analysis is performed using a custom-written MATLAB pipeline (available and deposited in https://github.com/JakubSalplachta/DUCT; *Hankeova, 2021* copy archived at swh:1:rev:6b0b0eb88bbaf9bfc4f8ee42ca-fa4c122866fbba) as well as with ImageJ.

Alagille syndrome (ALGS) is a congenital disorder affecting multiple organs, including the hepatic and cardiovascular systems (*Spinner et al., 1993*; *Mašek and Andersson, 2017*). The disease is usually caused by mutations in the Notch ligand *JAGGED1* (*JAG1*, OMIM: ALGS1 [*Oda et al., 1997*; *Li et al., 1997*; *Gilbert et al., 2019*]) or, less frequently, in the receptor *NOTCH2* (OMIM: ALGS2 [*Spinner et al., 1993*; *McDaniell et al., 2006*]), and is chiefly characterized by intrahepatic peripheral bile duct paucity (*Alagille et al., 1975*; *Riely et al., 1979*). Importantly, bile duct development is regulated by portal vein mesenchyme (*Hofmann et al., 2010*), implying that the architectural relationship between liver cells and the vasculature affects opportunities for signaling cross-talk between these systems in liver. Some patients with ALGS spontaneously recover a biliary system (*Riely et al., 1979*; *Fujisawa et al., 1994*). Ductular reaction, aberrant biliary growth and/or trans-differentiation from hepatocytes can contribute to biliary recovery in ALGS and other cholestatic disorders (*Schaub et al., 2018*; *Fabris et al., 2007*). Furthermore, it has been reported that liver vascular architecture is affected in ALGS (previously also known as arteriohepatic dysplasia or syndromic paucity of bile ducts [*Hadchouel et al., 1978*]). It is thus clear that understanding and defining both biliary and vascular intrahepatic defects is essential for ALGS.

DUCT is a versatile, reliable tool allowing standardized architecture analysis and definition of multiple lumenized trees on a whole organ level, facilitating systems insights. We demonstrated the applicability of DUCT by revealing the distinct morphological features that allow the de novo generated *Jag1*$^{Ndr/Ndr}$ adult biliary system to achieve wild-type biliary volume: (1) an increase in the number of central low generation branches and (2) profound tortuosity in the liver periphery. We confirmed these 3D findings in 2D sections from *Jag1*$^{Ndr/Ndr}$ mice and patients with ALGS, demonstrating that the new phenotypes identified with DUCT in the mouse model are representative of patient pathology. Using DUCT we also discovered novel phenotypes such as bile duct bridging between two portal veins, which would be misinterpreted as bile duct proliferation in 2D histological sections. Hence, 2D histological sections are not sufficient to understand the structural abnormalities of tubular networks.

## Results

### DUCT revealed that *Jag1*$^{Ndr/Ndr}$ adult mice generate a full-volume biliary system

In order to define and quantify the adaptive process resulting in a de novo generated biliary system in adult *Jag1*$^{Ndr/Ndr}$ mice, we investigated the spatial relationship of portal venous and biliary systems in normal and diseased liver in 3D. First, we compared double carbon ink injection (*Kaneko et al., 2015*) and whole mount immunofluorescence staining combined with tissue clearing using iDISCO+ (*Renier et al., 2016*) to assess the 3D architecture of the liver (*Figure 1—figure supplement 1A and B*). Neither carbon ink injection nor iDISCO+ (due to poor labeling of vascular network) were suitable for dual 3D analysis of vascular and biliary networks. We therefore developed an alternative approach for 3D analysis: DUCT (*Figure 1A*, *Figure 1—figure supplement 1C*) followed by semi-automated segmentation generating 3D binary masks of these two systems. The binary masks were used directly for DUCT data volume analysis. For further quantification of the architectural parameters the binary masks were skeletonized and analyzed (*Figure 1B*) using a custom-written MATLAB pipeline, ImageJ, or qualitative visual assessment.

DUCT outperformed ink injection and immunofluorescence in most aspects (*Figure 1—figure supplement 1D*), from 3D analysis (not possible with ink) to analysis of lumenization (not possible with immunofluorescence). One limitation, however, is that DUCT cannot visualize lumens smaller than 5 μm. Finally, to test whether DUCT can be applied to other organ systems, we visualized lung architecture by injecting the airways (via the trachea) and the vasculature (via the pulmonary artery) and 3D reconstructed the respiratory and vascular systems (*Figure 1—figure supplement 1E*). In lungs, as in liver, the two lumenized systems can be clearly distinguished using DUCT. In all the DUCT liver experiments in this manuscript, we applied DUCT to the right medial lobe, and used the other lobes for sample-matched quality control (*Figure 1—figure supplement 2*).

Our previous work revealed that *Jag1*$^{Ndr/Ndr}$ bi-potential hepatoblasts did not differentiate into cholangiocytes during embryonic development. Newborn *Jag1*$^{Ndr/Ndr}$ mice were jaundiced and displayed intrahepatic bile duct paucity. However, by adulthood *Jag1*$^{Ndr/Ndr}$ livers exhibited mature

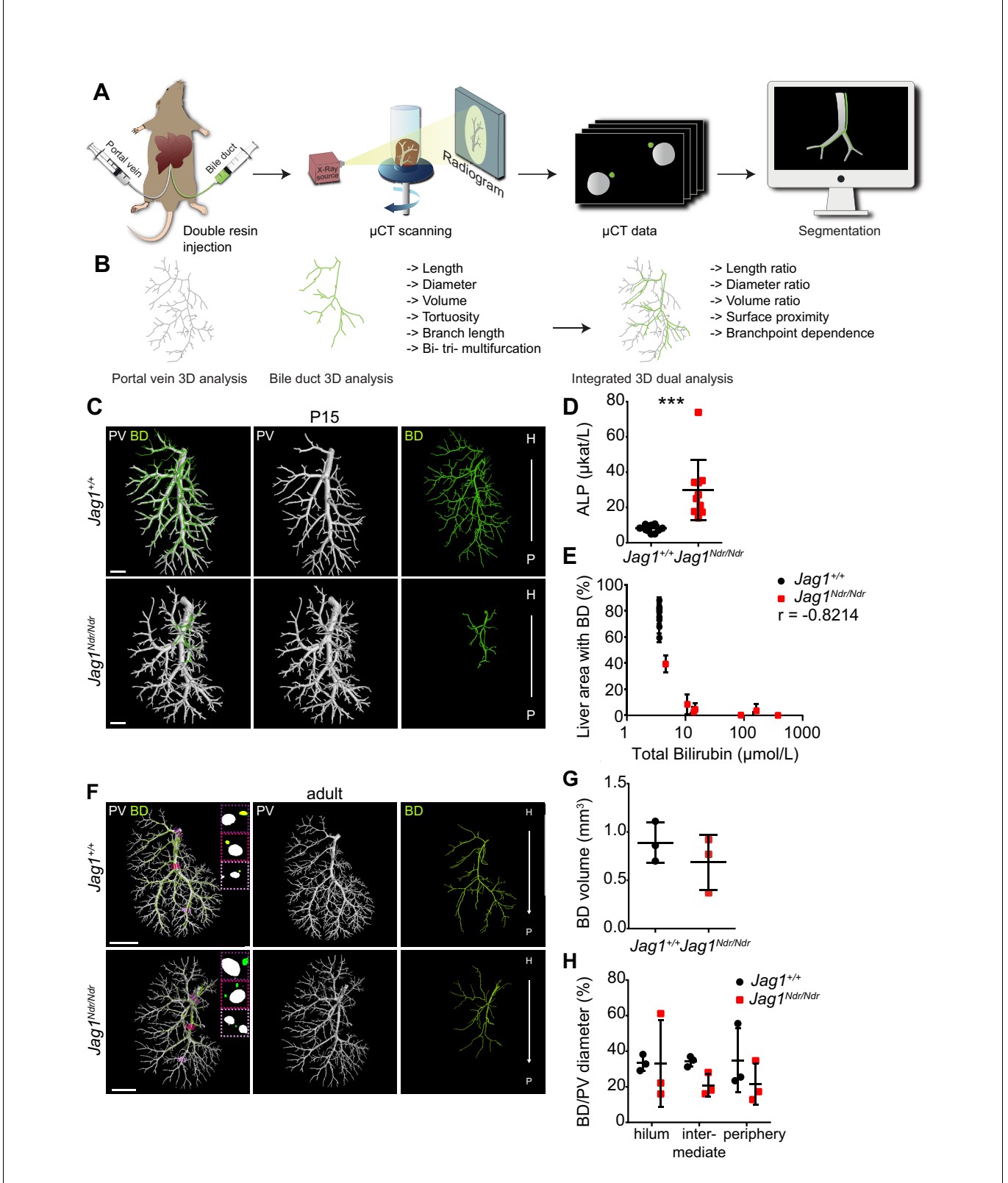

**Figure 1.** DUCT revealed that *Jag1^Ndr/Ndr* bile ducts recover a full-volume biliary system. (**A**) The DUCT pipeline encompasses resin injection into two systems (portal venous and biliary), micro computed tomography (μCT) scanning of the organ, or individual lobes, and segmentation of μCT data (tomographs) into 3D binary masks. (**B**) The image analysis pipeline creates 3D skeletons from the binary masks, to quantify architectural parameters in individual or combined systems. (**C**) 3D rendering of BD and PV structures using DUCT in postnatal day 15 (P15) *Jag1^+/+* (top panel) and *Jag1^Ndr/Ndr*

*Figure 1 continued on next page*

*Figure 1 continued*

livers (bottom panel). Scale bar = 1 mm. (**D**) Alkaline phosphatase (ALP) serum levels at P15. Each dot represents one animal; lines show mean value ± standard deviation. Statistical test – unpaired *t*-test, p=0.0008. (**E**) Correlation analysis between total bilirubin levels and liver area with resin-injected BD. Each dot represents one animal; lines show mean value ± standard deviation (measured in right medial and left lateral lobe). Statistical test – non-parametric Spearman correlation, p=0.0341, r = −0.8214. (**F**) 3D rendering of BD and PV structures using DUCT in adult (4.5–6.5 months old) *Jag1*$^{+/+}$ (top panel) and adult de novo generated *Jag1*$^{Ndr/Ndr}$ livers (bottom panel). Boxed regions highlight 2D sections of hilar, intermediate and peripheral regions identified with dotted lines in matched colors. Scale bar = 4 mm. (**G**) BD system volume is similar in adult *Jag1*$^{+/+}$ and *Jag1*$^{Ndr/Ndr}$ mice. Each dot represents one animal; lines show mean value ± standard deviation, unpaired *t*-test, p=0.3730 (**H**) BD/PV diameter ratio in *Jag1*$^{+/+}$ and *Jag1*$^{Ndr/Ndr}$ mice in hilar, intermediate and peripheral regions. Each dot represents one animal; lines show mean value ± standard deviation. Two-way ANOVA, p=0.2496. 3D, three dimensional; ALP, alkaline phosphatase; BD, bile duct; DUCT, double resin casting micro computed tomography; H, hilar; P, peripheral. PV, portal vein.

The online version of this article includes the following source data and figure supplement(s) for figure 1:

**Figure supplement 1.** DUCT outperforms other state of the art techniques to visualize mouse liver in 3D.

**Figure supplement 2.** Resin injection quality control of the left lateral lobe.

**Figure supplement 3.** Liver cast of P15 *Jag1*$^{+/+}$ showing bile duct (green) and portal vein (white) together (top panel) or separately (bottom panels).

**Figure supplement 3—source data 1.** 3D interactive liver cast shown in *Figure 1—figure supplement 3*.

**Figure supplement 4.** Liver cast of P15 *Jag1*$^{Ndr/Ndr}$ showing bile duct (green) and portal vein (white) together (top panel) or separately (bottom panels).

**Figure supplement 4—source data 1.** 3D interactive liver cast shown in *Figure 1—figure supplement 4* .

**Figure supplement 5.** *Jag1*$^{Ndr/Ndr}$ bile ducts displayed heterogeneous de novo growth.

**Figure supplement 6.** Liver cast of adult *Jag1*$^{+/+}$ showing bile duct (green) and portal vein (white) together (top panel) or separately (bottom panels).

**Figure supplement 6—source data 1.** 3D interactive liver cast shown in *Figure 1—figure supplement 6*.

**Figure supplement 7.** Liver cast of adult *Jag1*$^{+/+}$ showing bile duct (green) and portal vein (white) together (top panel) or separately (bottom panels).

**Figure supplement 7—source data 1.** 3D interactive liver cast shown in *Figure 1—figure supplement 7*.

**Figure supplement 8.** Liver cast of adult *Jag1*$^{+/+}$ showing bile duct (green) and portal vein (white) together (top panel) or separately (bottom panels).

**Figure supplement 8—source data 1.** 3D interactive liver cast shown in *Figure 1—figure supplement 8*.

**Figure supplement 9.** Liver cast of adult *Jag1*$^{Ndr/Ndr}$ showing bile duct (green) and portal vein (white) together (top panel) or separately (bottom panels).

**Figure supplement 9—source data 1.** 3D interactive liver cast shown in *Figure 1—figure supplement 9*.

**Figure supplement 10.** Liver cast of adult *Jag1*$^{Ndr/Ndr}$ showing bile duct (green) and portal vein (white) together (top panel) or separately (bottom panels).

**Figure supplement 10—source data 1.** 3D interactive liver cast shown in *Figure 1—figure supplement 10*.

**Figure supplement 11.** Liver cast of adult *Jag1*$^{Ndr/Ndr}$ showing bile duct (green) and portal vein (white) together (top panel) or separately (bottom panels).

**Figure supplement 11—source data 1.** 3D interactive liver cast shown in *Figure 1—figure supplement 11*.

**Figure supplement 12.** Overview of 3D reconstructed bile duct and portal vein systems, their branching skeletons and volume.

bile ducts, but with abnormal apical polarity (*Andersson et al., 2018*). Using DUCT, we first investigated to what degree *Jag1*$^{Ndr/Ndr}$ mice can grow a biliary tree by postnatal day 15 (P15). We examined the biliary system in 7 *Jag1*$^{Ndr/Ndr}$ P15 pups and discovered a rudimentary or absent biliary tree in *Jag1*$^{Ndr/Ndr}$ pups, while there was a fully developed biliary tree in *Jag1*$^{+/+}$ mice (*Figure 1C*). The 3D reconstruction of both systems can be explored in separate channels or in tandem with rotation (interactive PDFs, *Figure 1—figure supplements 3* and *4*). We noted high variability in biliary outgrowth between the right medial lobe (RML) and the left lateral lobe (LLL) in *Jag1*$^{Ndr/Ndr}$ livers (illustration of liver lobes *Figure 1—figure supplement 5A*). In the RML of five *Jag1*$^{Ndr/Ndr}$ pups, the biliary tree covered >5% of liver area, whereas in LLL only 1 *Jag1*$^{Ndr/Ndr}$ pup displayed >5% biliary tree coverage, while four *Jag1*$^{Ndr/Ndr}$ pups had no lumenized bile ducts. In *Jag1*$^{+/+}$ pups the biliary network covered on average 75% of the liver area in both RML and LLL (*Figure 1—figure supplement 5B and C*). The P15 *Jag1*$^{Ndr/Ndr}$ pups were cholestatic, and manifested increased levels of alkaline phosphatase (ALP) (*Figure 1D*), aspartate aminotransferase (AST), alanine aminotransferase (ALT) and decreased levels of albumin (*Figure 1—figure supplement 5D*). Interestingly, two different groups were noted in *Jag1*$^{Ndr/Ndr}$ pups with regard to the total bilirubin levels, that is, 50% of the animals had highly increased and 50% had mildly increased total bilirubin (p=0.0079), while all *Jag1*$^{+/+}$ mice displayed bilirubin levels below detection limits (*Figure 1—figure supplement 5D*). We correlated the total bilirubin amount with the liver area covered by lumenized bile ducts and detected a strong negative correlation between these two factors (*Figure 1E*, r = −0.8214). We further sectioned the resin injected P15 liver and stained for the early biliary marker SOX9 (SRY-Box

transcription factor 9) (note remaining resin in some portal veins). In $Jag1^{+/+}$ central liver, lumenized bile ducts were clearly detected, whereas in all four $Jag1^{Ndr/Ndr}$ central livers no lumenized bile ducts were visible and the number of SOX9 positive cells varied between animals, poorly reflecting the total bilirubin levels (*Figure 1—figure supplement 5E*).

Next, we aimed to characterize and quantify the biliary architecture of the $Jag1^{Ndr/Ndr}$ de novo generated biliary system in adult mice (*Figure 1F*). The 3D reconstruction of both systems can be explored in separate channels or in tandem (interactive PDFs, *Figure 1—figure supplement 6– 11*). While $Jag1^{+/+}$ mice demonstrated a stereotyped vascular and biliary architecture (*Figure 1—figure supplement 12A, B and C*), $Jag1^{Ndr/Ndr}$ livers exhibited greater architectural variability (*Figure 1—figure supplement 12D, E and F*). To quantify the degree of de novo biliary formation, we extracted the volume (using the binary masks) and diameters of the portal venous and biliary systems (using the binary masks and skeletons). The total volume of the vascular and biliary trees was similar in $Jag1^{Ndr/Ndr}$ and $Jag1^{+/+}$ mice (*Figure 1G*, *Figure 1—figure supplement 12H*). There was a tendency toward a larger portal venous volume and smaller biliary volume in $Jag1^{Ndr/Ndr}$ mice, resulting in a trend toward a reduced BD:PV (bile duct, portal vein) volume ratio in $Jag1^{Ndr/Ndr}$ mice (*Figure 1—figure supplement 12H*, p=0.0594). Next, we investigated portal vein and bile duct diameters along the main branch. There was a tendency to an increase in the $Jag1^{Ndr/Ndr}$ portal vein diameter and a decrease in biliary diameter as a function of distance compared with $Jag1^{+/+}$ mice (*Figure 1—figure supplement 12I*). However, there was high variability in venous and bile duct diameter in the $Jag1^{Ndr/Ndr}$ hilar region. In $Jag1^{+/+}$ liver, the BD:PV diameter ratio was consistently 1:3 in hilar and intermediate regions (*Figure 1H*). This BD:PV diameter ratio was not preserved in the $Jag1^{Ndr/Ndr}$ livers (*Figure 1H*).

DUCT enabled 3D visualization of postnatal and adult biliary and vascular trees. The biliary network in $Jag1^{Ndr/Ndr}$ mice appeared postnatally, but with high heterogeneity between liver lobes and animals. Moreover, the liver area with lumenized bile ducts correlated with total bilirubin levels and disease severity at P15. The adult segmented μCT data were analyzed for volume and inner diameter of two injected systems. Comparisons of the portal vein and bile duct diameters demonstrated a conserved portal vein – bile duct architectural relationship with a stereotype BD:PV 1:3 diameter ratio in $Jag1^{+/+}$ liver. The adult $Jag1^{Ndr/Ndr}$ mice displayed a heterogeneous phenotype that nevertheless resulted in full restoration of biliary function (*Andersson et al., 2018*) via recovery of a wild-type biliary volume.

## Alagille syndrome human and murine bile ducts end abruptly

The intrahepatic biliary tree forms by a tubulogenic process in which a heterogeneous, hierarchical fine mesh of connected cholangiocytes is refined to single larger conduits of bile ducts, resembling a branching tree (*Ober and Lemaigre, 2018*; *Tanimizu et al., 2016*). In $Jag1^{+/+}$ liver, the qualitative analysis of network connectivity revealed that the biliary system formed a continuous tree, branching outwards toward the periphery as expected (*Figures 1C, F* and *2A* left panel). In contrast, the $Jag1^{Ndr/Ndr}$ biliary system displayed some branches oriented from peripheral to hilar, with abrupt endings (*Figure 2A* right panels, blue arrowheads, on average one abruptly ending BD per lobe). We confirmed the abruptly ending bile ducts in $Jag1^{Ndr/Ndr}$ livers in serial liver sections (*Figure 2B*, bottom panels). The black arrowhead labels a well-formed bile duct, that ended bluntly in the following section (+5 μm, blue arrowhead), and disappeared completely in the next section (+10 μm).

To determine whether the $Jag1^{Ndr/Ndr}$ biliary abnormalities were representative of pathology in patients with Alagille syndrome, we evaluated liver serial sections from whole liver explants (patients with severe Alagille syndrome (S-ALGS) that underwent transplantation) and biopsies obtained for clinical reasons in non-transplanted patients (patients with mild Alagille syndrome (M-ALGS)).The liver function tests for individual patients are reported in *Table 1* and representative liver sections are presented in *Figure 2—figure supplement 1*. One patient (M_ALGS-5) had been biopsied at multiple time points revealing paucity of bile duct at 2.5 months, a regenerating phase with hepatocytes expressing CK7 at 1.3 years and bile duct recovery at 4.7 years (*Figure 2—figure supplement 1D*).

We evaluated liver sections from patients with severe (top panel) and mild (bottom panel) Alagille syndrome for the presence of abruptly terminating bile ducts. A black arrowhead labels a well-formed bile duct that terminated in the subsequent section (+5 μm, blue arrowhead), (*Figure 2C*). In conclusion, DUCT facilitated qualitative assessments of the tubular networks, including connectivity

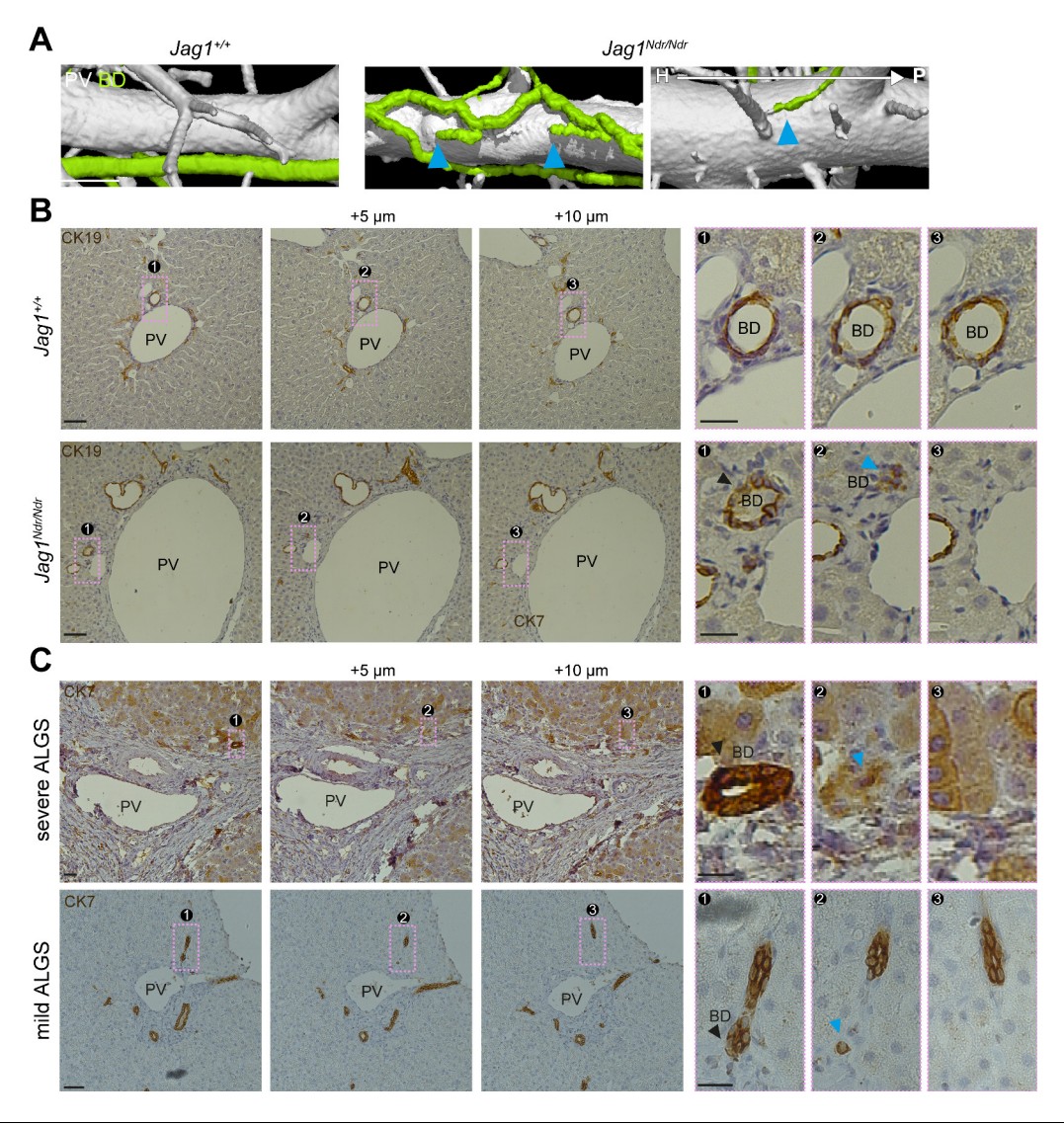

**Figure 2.** Alagille syndrome human and murine bile ducts end abruptly. (**A**) *Jag1^{Ndr/Ndr}* BDs (right panel) terminated randomly and facing toward the hilum (blue arrowheads). (**B**) 2D histological consecutive liver sections confirmed abruptly terminating BDs in *Jag1^{Ndr/Ndr}* liver. Black arrowhead depicts lumenized well-formed BD that ended in the following sections (blue arrowhead). (**C**) BDs in patients with severe ALGS (top panel) or mild ALGS (bottom panel) terminate abruptly (blue arrowhead) in consecutive liver histological sections. Scale bars (**A**) 500 µm, (**B** left panels), (**C**) 50 µm, (**B** boxed region) 20 µm. ALGS, Alagille syndrome; BD, bile duct; CK, cytokeratin; H, hilar; P, peripheral. PV, portal vein.

The online version of this article includes the following figure supplement(s) for figure 2:

**Figure supplement 1.** Overview of liver samples from patients with Alagille syndrome stained for CK7.

and perfusion. Both the *Jag1^{Ndr/Ndr}* and Alagille syndrome biliary systems displayed abruptly ending bile ducts, which may affect bile flow and shear stress.

## Alagille syndrome human and murine de novo generated bile ducts are further from portal veins

During embryonic development, cholangiocytes differentiate from hepatoblasts that are in contact with portal vein mesenchyme expressing *Jag1* (*Ober and Lemaigre, 2018*). Whether postnatally de novo generated bile ducts arise adjacent to the portal vein, or whether they are less dependent on portal vein proximity has not yet been explored. We therefore analyzed the distance between the biliary and portal venous systems by calculating the surface distances using the MATLAB pipeline.

**Table 1.** Liver function test overview for patients with Alagille syndrome.
M-ALGS stands for patients with mild Alagille syndrome. S-ALGS stands for patients with severe Alagille syndrome.

| ID | Age (years) | ALT (μkat/L) | AST (μkat/L) | ALP (μkat/L) | GT (μkat/L) | Bil Tot (μmol/L) | BilD (μmol/L) | Bile acids (μmol/L) | Sample |
|---|---|---|---|---|---|---|---|---|---|
| Reference value | | <0.76 | <1 | <7.6 | <0.76 | <22 | <4 | <10 | |
| S-ALGS_1 | 1.2 | 3.29 | 2.87 | 23.4 | 21.8 | 308 | 272 | 665 | Explant |
| S-ALGS_2 | 1.5 | 3.09 | - | 10.8 | 13.5 | 274 | 246 | 473 | Explant |
| S-ALGS_3 | 4 | 4.41 | - | 9.9 | 7.5 | 238 | 215 | 246 | Explant |
| S-ALGS_4 | 6 | 3.9 | - | 9.4 | 8.1 | 220 | 193 | 244 | Explant |
| M-ALGS_1 | 0.6 | 1.09 | - | 4.6 | 1.2 | 7 | 2 | - | Biopsy |
| M-ALGS_2 | 1.4 | 0.78 | 0.96 | 6.6 | 1.2 | 3 | <2 | 33 | Biopsy |
| M-ALGS_3 | 2.5 | 1.46 | 1.44 | 13.6 | 7.4 | 10 | 6 | 106 | Biopsy |
| M-ALGS_4 | 3.4 | 0.68 | 0.92 | 5.2 | 1.5 | 14 | 4 | 31 | Biopsy |
| M-ALGS_5_1 | 0.2 | 1.44 | 1.77 | - | - | 46 | 39 | 175 | Biopsy |
| M-ALGS_5_2 | 1.2 | 1.59 | 1.8 | - | - | 3 | 2 | - | Biopsy |
| M-ALGS_5_3 | 4.7 | 0.79 | 0.93 | 3.5 | 0.52 | 9 | <2 | 2 | Biopsy |
| M-ALGS_6 | 6.4 | 1.18 | 1.44 | 6.4 | 3 | 6 | <2 | 1 | Biopsy |

Specifically, the surface distance was defined as the shortest length from biliary skeleton to the portal venous skeleton, minus radiuses of these systems at the defined points (*Figure 3A*). *Jag1$^{+/+}$* bile ducts maintained a uniform distance to adjacent portal veins throughout the liver (*Figure 3B* top panel, asterisk, 3C, 3D 100% of BDs within 0.5 mm of a PV). In contrast, *Jag1$^{Ndr/Ndr}$* bile ducts did not maintain a uniform distance to the nearest portal vein (*Figure 3B* bottom panel, double arrow, 3C, 3D 1.5% of BD are placed 0.5–1.26 mm away from a PV) and sometimes traversed the parenchyma to join another portal vein branch (*Figure 3B* bottom right panel, *Figure 1—figure supplement 2* empty arrowheads). Both the increased BD-PV distance and parenchymal bile ducts were validated in histological sections, in which *Jag1$^{+/+}$* bile ducts were in close proximity to, or embedded in portal vein mesenchyme (*Figure 3E* left panel, asterisk). In contrast, *Jag1$^{Ndr/Ndr}$* bile ducts were confirmed to be present outside of the portal vein mesenchyme area (*Figure 3E* middle panel, double arrow) or even in the liver parenchyma close to the edge of the liver (*Figure 3E* right panel). This phenotype, visualized in 2D sections, could resemble biliary proliferation, or ductular reaction, rather than a bridging structure, highlighting the importance of 3D imaging. Parenchymal bile ducts were also detected in liver samples from patients with severe and mild Alagille syndrome (*Figure 3F*) but not in control human liver. In sum, DUCT pipeline together with the MATLAB algorithm, measured the gap between surfaces of two resin injected systems to address the spatial relationship between them. Our data showing biliary cells in the parenchyma and bile ducts far from portal veins in *Jag1$^{Ndr/Ndr}$* liver and liver from patients with Alagille syndrome thus suggest that postnatal bile duct formation does not rely on close proximity to portal vein mesenchyme and may occur independent of signals from portal vein mesenchyme.

## Alagille syndrome human and murine de novo generated bile ducts display branching independent of portal vein branching

Portal vascular and biliary systems are ductal tree-like structures with numerous branches, which function to maximize the area of exchange between the tissue and its lumen. We evaluated portal venous and biliary branching using the DUCT pipeline and the MATLAB script to quantify the total number of vascular or biliary branch points. Branch points were identified using the 3D skeletons of each system, and categorized based on the number of incoming/outgoing branches (classifying as bifurcations, trifurcations, or nodes with more than three branches) (*Figure 1—figure supplement 12D and G*). We did not identify any differences in the absolute numbers of branch points in *Jag1$^{+/+}$* and *Jag1$^{Ndr/Ndr}$* systems, again suggesting that de novo biliary growth generally reconstituted a full-volume, well-branched biliary tree (*Figure 4—figure supplement 1*).

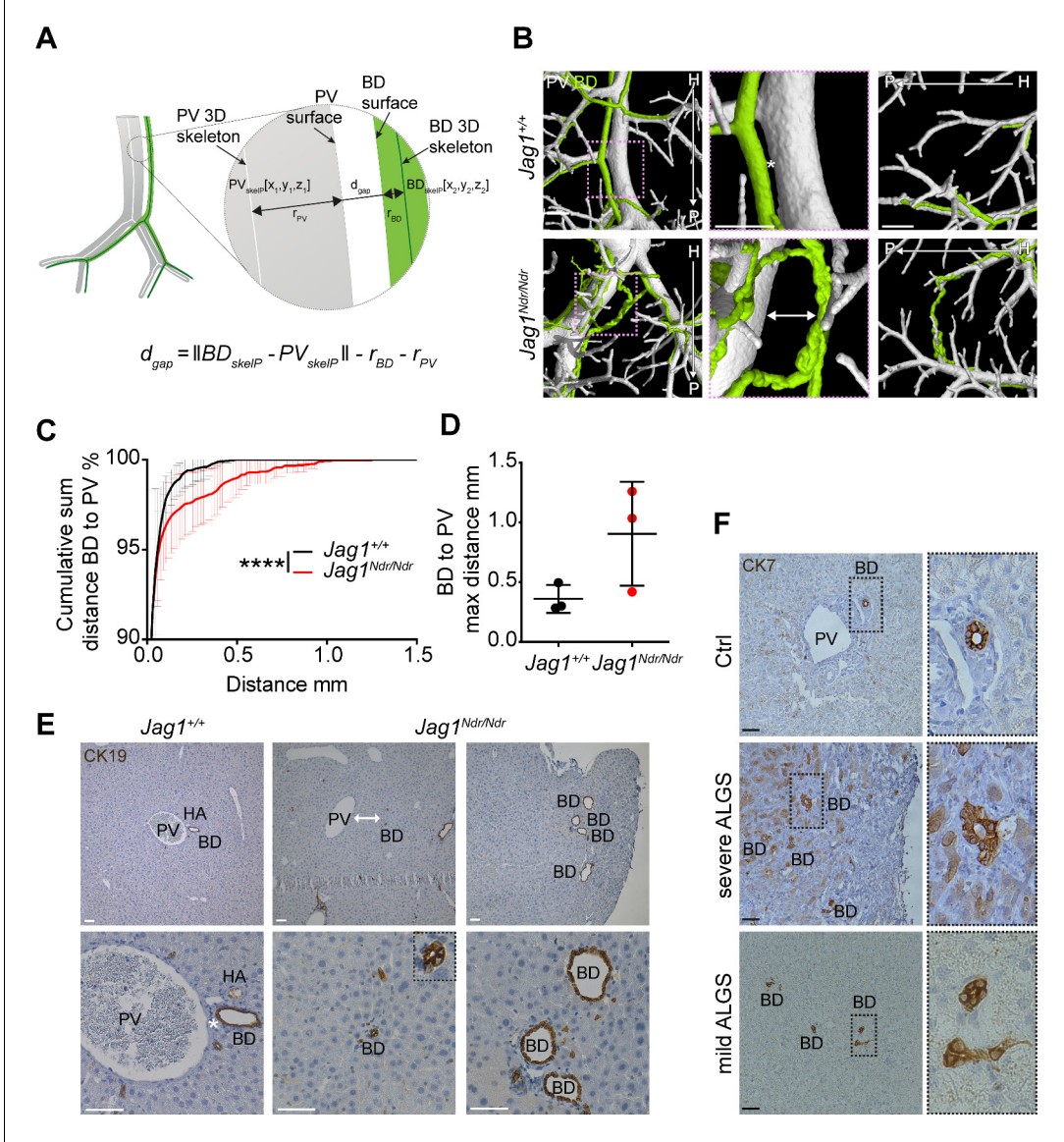

**Figure 3.** Alagille syndrome human and murine de novo generated bile ducts are further from portal veins. (**A**) Scheme of BD to PV surface distance analysis. $PV_{skelP}$ = single point on PV skeleton, $BD_{skelP}$ = single point on BD skeleton, $r_{PV}$ = radius of PV at $PV_{skelP}$ (i.e. minimal distance from $PV_{skelP}$ to PV surface), $r_{BD}$ = radius of BD at $BD_{skelP}$ (i.e. minimal distance from $BD_{skelP}$ to BD surface), $d_{gap}$ = gap distance, which is derived by subtracting the radii from the skeleton to skeleton distance. (**B**) 3D rendering shows homogenous distance between a BD and PV in $Jag1^{+/+}$ livers (asterisk), but a large heterogeneous distance in $Jag1^{Ndr/Ndr}$ liver (double-headed arrow). Right panel shows a parenchymal bile duct traversing between two PVs at the $Jag1^{Ndr/Ndr}$ liver edge. Scale bar 500 µm. (**C**) Cumulative sum of percentage of BDs at a given distance from the nearest PV. 3 $Jag1^{+/+}$ and 3 $Jag1^{Ndr/Ndr}$ mice were used. Bars represent mean ± standard deviation, Kolmogorov - Smirnov test (on raw data), p<0.0001, (****). For individual data points see *Figure 3—source data 1*. (**D**) Maximum distance between BD and PV. Each dot represents one animal, bars are mean ± standard deviation, unpaired *t*-test, p=0.1041, not significant. (**E**) BD – PV distances confirmed in 2D histological sections. Overview in top panel and magnification in bottom panel. $Jag1^{Ndr/Ndr}$ PVs can be present in the parenchyma far from (middle panels), or independent of (right panels), the nearest PV. Scale bars 50 µm. (**F**) Healthy human liver with BD close to PV (top panel). Parenchymal CK7+ BDs in histological liver sections from a patient with severe ALGS (middle panel) and distant BDs in a patient with mild ALGS (bottom panel). Magnification shows lumenization of BDs. Scale bar 50 µm. ALGS, Alagille syndrome, BD, bile duct; CK, cytokeratin; H, hilar; HA, hepatic artery; P, peripheral; PV, portal vein.

The online version of this article includes the following source data for figure 3:

**Source data 1.** Raw data measuring the distance from the surface of bile duct to a portal vein surface.

During embryonic development, the biliary system is established alongside the portal venous system. This is reflected in the final architecture of the system, with bile ducts in close proximity to portal veins (*Figure 3*). A prediction based on this embryonic process and BD/PV dependency is therefore that bile ducts should invariably branch where portal veins branch. We extracted the coordinates for branch points in the biliary and portal venous systems from the corresponding 3D skeletonized data and calculated 3D Euclidean distances between biliary branch points and their nearest neighboring portal vein branch point (*Figure 4A*). Indeed, $Jag1^{+/+}$ bile ducts (*Figure 4B* left panel, blue arrowhead) branched adjacent to portal vein branch points (magenta arrowhead, defined as within 0.5 mm). In contrast, $Jag1^{Ndr/Ndr}$ bile ducts (*Figure 4B* middle panel, blue arrowhead) branched further from portal vein branch points (magenta arrowhead), or independent of portal vein branch points (blue arrow in *Figure 4B*, and data in 4C; independence defined as distance >0.54 mm). On average, 1.3% of $Jag1^{+/+}$ bile ducts branch points and 8.6% of $Jag1^{Ndr/Ndr}$ bile duct branch points were further than 0.5 mm from the nearest portal vein branch point (*Figure 4C*). We analyzed consecutive histological liver sections to confirm branching morphology defects discovered using DUCT. $Jag1^{+/+}$ bile ducts (*Figure 4D* top panel, blue arrowhead) indeed branched at the same point as portal veins branch (magenta arrowhead). In contrast, in $Jag1^{Ndr/Ndr}$ liver bile ducts might bifurcate in the absence of portal vein branching (*Figure 4D* bottom panel, blue arrow).

We next asked whether similar branching phenotypes were present in healthy human liver or in patients with mild or severe Alagille syndrome. In normal human liver, biliary branching occurred close to portal vein bifurcation (*Figure 4E* top panel, blue arrowheads, PV branching within 25 µm in this example, branching not shown). In patients with severe Alagille syndrome, biliary branching could be seen independent of portal vein branching (*Figure 4E* middle panel, blue arrows), the nearest portal vein branch point for this bile duct was 13 sections hilar (circa 65 µm earlier). We also detected independent bile duct branching in patients with mild Alagille syndrome (*Figure 4E* bottom panel, blue arrows). In conclusion, DUCT revealed dual system 3D architectural phenotypes: (1) similar numbers of branch points in $Jag1^{+/+}$ and $Jag1^{Ndr/Ndr}$ livers but (2) a greater distance between portal vein and bile duct branch points in $Jag1^{Ndr/Ndr}$ livers. Bile ducts in patients with Alagille syndrome displayed similar branching abnormalities, corroborating the architectural independence from portal vein patterning.

## Strahler analysis of resin casts reveal excess central branching in the $Jag1^{Ndr/Ndr}$ de novo generated biliary system

Whether de novo bile duct formation occurs evenly throughout the liver or is more extensive in certain regions has not yet been quantitatively defined. One of the mechanisms by which the biliary system can regenerate is via abundant branching of the network (*Vartak et al., 2016*; *Masyuk et al., 2001*). We therefore employed the DUCT pipeline and ImageJ to perform a Strahler analysis (3D branching analysis based on generation number from the origin, see Material and methods section 'Branching analysis' for details on generation number calculation) to address the branch length, number and distribution in specific liver areas. In order to define anatomy correctly for differently sized livers, the liver lobe data were separated into three equivalent regions using portal vein branch generation number and lengths as a proxy for hilum/intermediate/periphery. These regions were denoted region 1 (R1 enriched for hilar region), 2 (R2, intermediate), and 3 (R3, peripheral-enriched) (*Figure 5A*). The average branch length of the portal vein was shorter in $Jag1^{Ndr/Ndr}$ livers, resulting in significantly smaller portal vein region sizes (*Figure 5B* middle panel), reflecting the overall smaller size of the mice and livers. Using the regions defined by the portal venous system, biliary region size was similar in the 1st and 2nd region of wild type and $Jag1^{Ndr/Ndr}$ mice, while region three was smaller in both animal groups due to resin not penetrating ducts < 5 µm (*Figure 5B* right panel, *Figure 5—figure supplement 1*). Two out of three $Jag1^{Ndr/Ndr}$ mice had a marked reduction in biliary region size in R3, while one had an increase, reflecting the somewhat variable phenotype manifested by both the patients and the mouse model.

Branching trees in biological systems have a stereotype structure in which branch lengths shorten with each branching generation from start (R1) to end (periphery, R3) (*Masyuk et al., 2001*; *Hannezo et al., 2017*). We analyzed the portal vein and biliary branch lengths within each region and found that portal vein segments shortened as expected with each generation in both $Jag1^{+/+}$ and $Jag1^{Ndr/Ndr}$ livers (*Figure 5C* middle panel). $Jag1^{+/+}$ bile ducts followed the same stereotype branching principle, but $Jag1^{Ndr/Ndr}$ bile ducts branch lengths were significantly shorter in regions 1

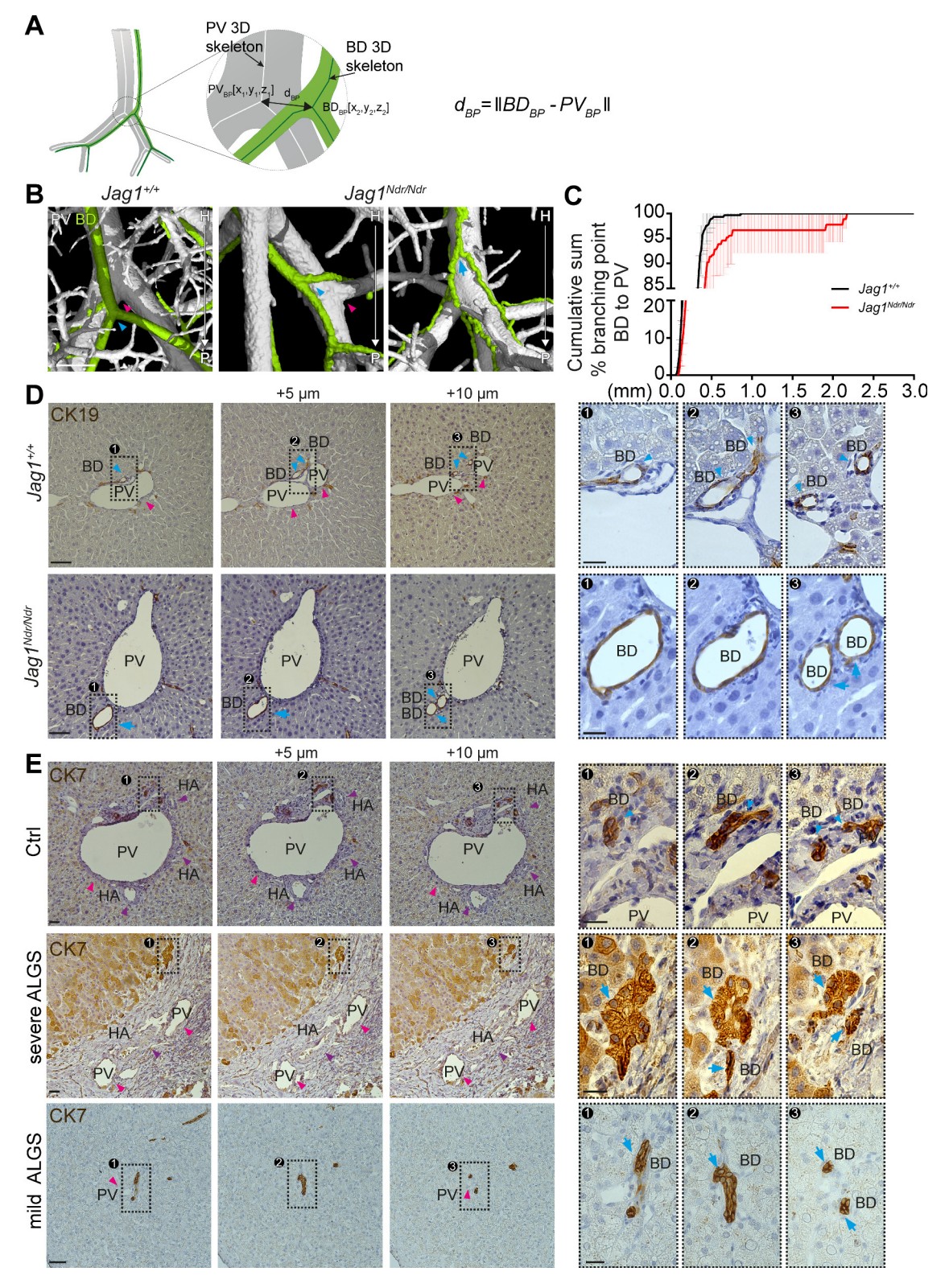

**Figure 4.** Alagille syndrome human and murine de novo generated bile ducts display branching independent of portal vein branching. (**A**) Scheme representing BD to PV branch point analysis. $PV_{BD}$ = PV branch point, $BD_{BD}$ = BD branch point, $d_{BP}$ = Euclidean 3D distance between branch points. (**B**) Branching pattern in $Jag1^{+/+}$ (left panel) and $Jag1^{Ndr/Ndr}$ liver (middle and right panel). PV branch points (pink arrowheads) were near BD branch points (blue arrowheads) in wild type mice, but further away in $Jag1^{Ndr/Ndr}$ mice. BD branch points in $Jag1^{Ndr/Ndr}$ mice also occurred in the absence of

*Figure 4 continued on next page*

*Figure 4 continued*

PV branching (blue arrow). Scale bar 500 µm. (C) Cumulative sum of BD branching point percentage at a given distance to the nearest PV branching point. 3 *Jag1*$^{+/+}$ and 3 *Jag1*$^{Ndr/Ndr}$ mice were used. 100% of *Jag1*$^{+/+}$ bile duct branchpoints were within 1 mm of a PV branchpoint, but only 95% of *Jag1*$^{Ndr/Ndr}$ branchpoints were within 1 mm. Bars represent mean ± standard deviation, Kolmogorov-Smirnov test (on raw data, p=0.9985, not significant). For individual data points see *Figure 4—source data 1* (D) Branching analysis in 2D histological consecutive *Jag1*$^{+/+}$ and *Jag1*$^{Ndr/Ndr}$ liver sections. PV branching (pink arrowheads) was present near BD branching (blue arrowheads) in wild type mice (top panels). In *Jag1*$^{Ndr/Ndr}$ mice, BDs branched ectopically in the absence of PV branching (bottom panel). Boxed regions magnified in panels at right. Scale bar 50 µm, boxed region 20 µm. (E) Branching pattern in consecutive human liver histological sections shows BD branching in association with PV (pink arrowhead) branching in controls (top panel), but BD branching in the absence of PV branching in patients with severe ALGS (middle panel) and patients with mild ALGS (bottom panel). Scale bar 50 µm, boxed region 20 µm. ALGS, Alagille syndrome; BD, bile duct; CK, cytokeratin; HA, hepatic artery; H, hilar; P, peripheral; PV, portal vein.

The online version of this article includes the following source data and figure supplement(s) for figure 4:

**Source data 1.** Raw data measuring the distance from the bile duct branching point to a portal vein branching point.

**Figure supplement 1.** De novo grown bile ducts did not show significant differences in the numbers of bifurcations, trifurcations or nodes > 3 branches when normalized to system size.

and 2, and uniform across the hierarchy of branches (*Figure 5C* right panel). We further analyzed the distribution of number of branches in the three regions and discovered that, on average, 9% of *Jag1*$^{+/+}$ portal vein branches fell into R1, 66% into R2% and 25% into R3, with a similar distribution in *Jag1*$^{Ndr/Ndr}$ livers (*Figure 5D* middle panel). Based on biological tree structure, we would expect fewest biliary branches in R1, an intermediate number in R2, and most branches in R3. However, resin penetration and lumen diameter precluded filling of all terminal portal vein branches in R3. Mirroring the wild type portal venous branch distribution, *Jag1*$^{+/+}$ biliary branch distribution was, on average, 16% in R1, 66% in R2% and 18% in R3. *Jag1*$^{Ndr/Ndr}$ biliary branch distribution was shifted, with 44% in R1, 44% in R2% and 12% in R3 (*Figure 5D* right panel). The distribution of biliary branches number was highly heterogeneous among the different *Jag1*$^{Ndr/Ndr}$ mice. DUCT pipeline is compatible with multiple image analysis software programs. Here, we used ImageJ to address the branching length over branching generations, and our data collectively indicated that low-generation number biliary segments were shorter and more numerous than expected, suggesting that ectopic regenerative branching and/or incorporation of de novo generated biliary cells, occurs in region 1. However, peripheral branching, which may be undetectable if biliary diameters are under 5 µm, cannot be excluded.

## Alagille syndrome human and murine de novo generated bile ducts are tortuous

The biliary system can adapt to excessive amounts of bile by increasing its diameter or length (*Vartak et al., 2016*; *Slott et al., 1990*). One means of enlarging length is by duct convolution. We therefore investigated the length and tortuosity of the biliary and portal vascular trees using the DUCT pipeline in combination with the MATLAB algorithm (whole system and main branch quantification) and ImageJ (R1, R2 and R3 analysis). 3D reconstruction of portal vein vasculature and the biliary network revealed straight *Jag1*$^{+/+}$ bile ducts (*Figure 6A* top panel), whereas *Jag1*$^{Ndr/Ndr}$ bile ducts were tortuous (*Figure 6A* bottom panel), especially in the liver periphery. We confirmed in histological liver sections that the *Jag1*$^{Ndr/Ndr}$ BDs were tortuous (*Figure 6B*). We further assessed biliary tortuosity in patients with mild Alagille syndrome and found several tortuous bile ducts (*Figure 6C*); however, we did not detect any tortuous bile ducts in patients with severe Alagille syndrome (data not shown). In order to quantify tortuosity, we calculated the actual (curved) length and theoretical (chord) length (scheme *Figure 6D*). The curved and chord lengths of the entire system, and the main branch alone did not differ for portal venous or biliary systems in *Jag1*$^{+/+}$ and *Jag1*$^{Ndr/Ndr}$ mice (*Figure 6—figure supplement 1A–G*). The BD:PV ratio was not significantly different for curved (*Figure 6—figure supplement 1H*) or chord length (*Figure 6—figure supplement 1I*). However, there was a 6% increase in overall portal venous tortuosity in *Jag1*$^{Ndr/Ndr}$ mice when the entire system was taken into account (*Figure 6E*) and biliary tortuosity was increased by 50%. Biliary tortuosity was greatest in the *Jag1*$^{Ndr/Ndr}$ liver periphery, with a 140% increase in R3 (*Figure 6E*). The tortuosity of the main portal vein or main bile ducts branch analyzed alone was not significantly different (*Figure 6—figure supplement 1J and K*), highlighting the importance of analyzing the

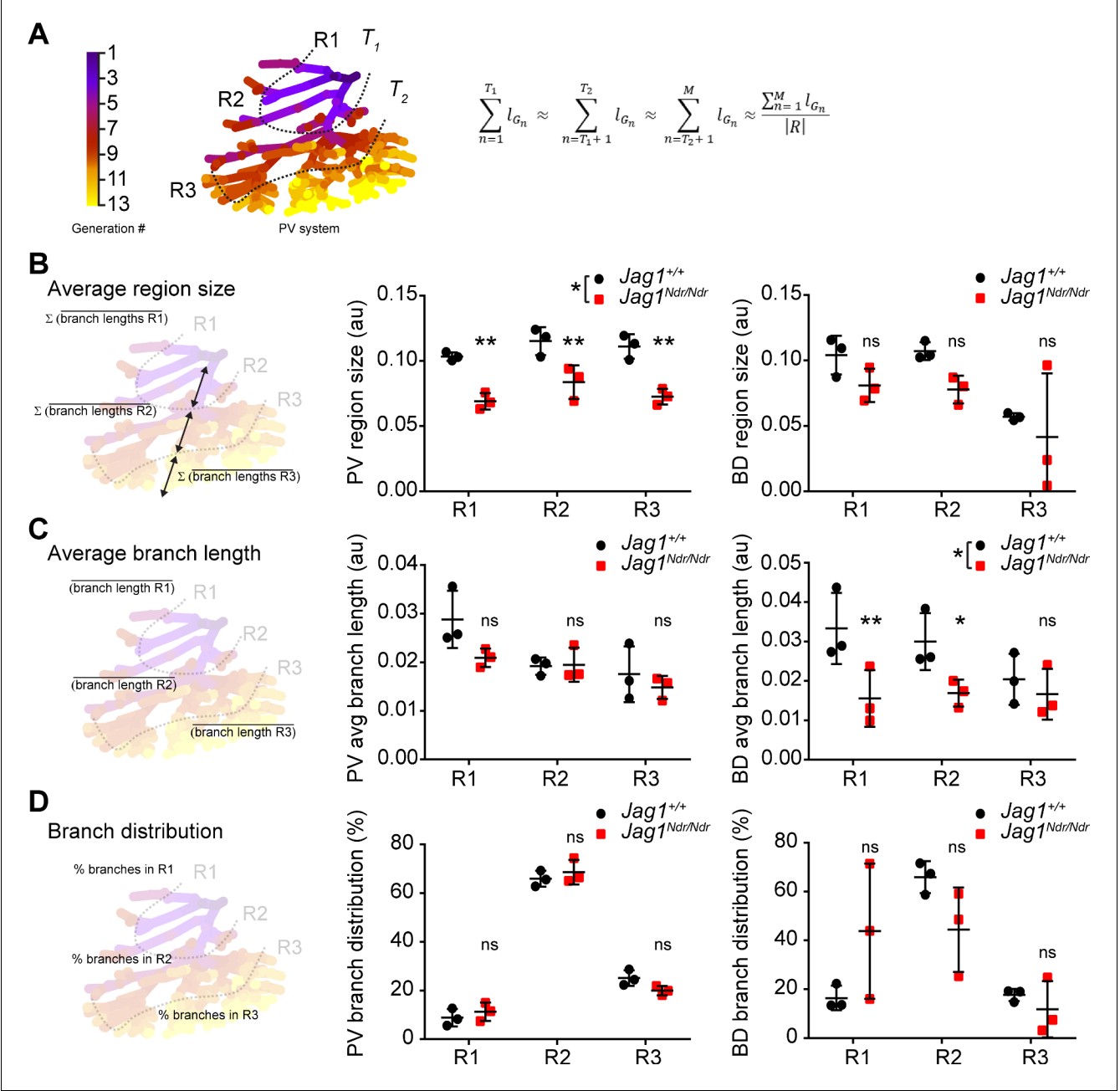

**Figure 5.** Strahler analysis of resin casts reveal excess central branching in the *Jag1^Ndr/Ndr* de novo generated biliary system. (**A**) 3D branching analysis based on Strahler number. The branching generations were divided into three equal regions: R1, R2 and R3 based on portal vein average branch length and generation number. Formula: R = {R1, R2, R3}; n ∈ [1, M] ⊂ ℕ; T$_1$, T$_2$⊂ n; R, region; G, branch generation; M, maximal branch generation number; l$_{Gn}$, average branch length of n$^{th}$ generation; T$_1$, T$_2$, borders between regions (specific generation number). (**B**) Schematic representation of region size calculation, deriving the sum of the average branch lengths within a given region (left). Liver region size for PV (middle panel) and BD (right panel). (**C**) Schematic representation of average branch length calculation within a region (left panel). Average branch length analysis for PV (middle panel) and BD (right panel) system. (**D**) Schematic representation of branch distribution, deriving the percentage of branches belonging to each region (left panel). Percentage of branches in each liver region in PV (middle panel) and BD (right panel) systems. Each dot represents one animal, bars represent mean ± standard deviation. Two-way ANOVA, (**B**) middle panel p = 0.0289, right panel p = 0.2029; (**C**) middle panel p = 0.1177, right panel p = 0.0367; (**D**) middle panel p = 0.4226, right panel p = 0.8845; followed by Sidak's multiple comparisons test, p < 0.05 (*), p < 0.01 (**), ns not significant. au, arbitrary units; BD, bile duct; PV, portal vein.

The online version of this article includes the following figure supplement(s) for figure 5:

**Figure supplement 1.** Bile duct branch distribution per region.

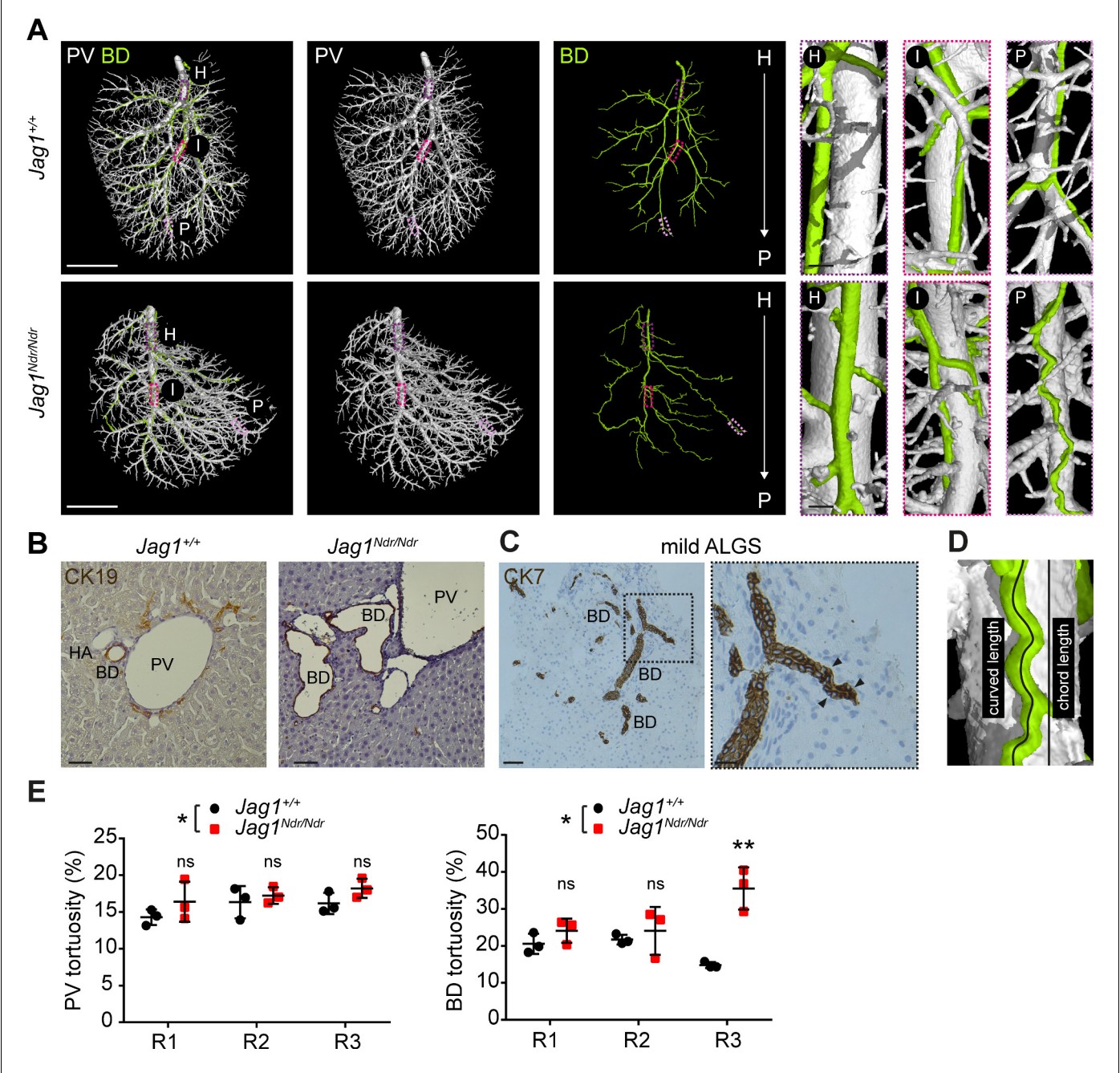

**Figure 6.** Alagille syndrome human and murine de novo generated bile ducts are tortuous. (**A**) DUCT 3D rendering of BD and PV structures in *Jag1*$^{+/+}$ (top panels) and *Jag1*$^{Ndr/Ndr}$ liver (bottom panels). Boxed areas magnify the hilar (**H**), intermediate (**I**) and peripheral (**P**) regions. Scale bars left 4 mm, boxed regions 250 μm. (**B**) 2D histological liver sections, show well-formed round CK19+ BDs in *Jag1*$^{+/+}$ liver, but aberrantly formed BDs in *Jag1*$^{Ndr/Ndr}$ liver. Scale bar 50 μm. (**C**) 2D liver section from patient with mild ALGS revealed tortuous misshaped BDs. Scale bar 50 μm, boxed region 20 μm. (**D**) Schematic representing length measurements. Percentage tortuosity was calculated by dividing curved (actual) length by chord (theoretical) length, and subtracting 100% (final value of 0% = not tortuous, perfectly straight). (**E**) The overall *Jag1*$^{Ndr/Ndr}$ PV (left graph) and BD (right graph) systems are more tortuous than wild types, and the *Jag1*$^{Ndr/Ndr}$ BD system is particularly tortuous in Region 3 (periphery). Each dot represents one animal, lines show mean value ± standard deviation. Statistical test: two-way ANOVA, left panel p=0.0141, right panel p=0.0251; followed by Sidak's multiple comparisons test; p<0.05 (*), p<0.01 (**). ALGS, Alagille syndrome; BD, bile duct; CK, cytokeratin; DUCT, double resin casting micro computed tomography, H, hilar; HA, hepatic artery, I, intermediate P, peripheral; PV, portal vein.

The online version of this article includes the following figure supplement(s) for figure 6:

**Figure supplement 1.** The adult de novo formed *Jag1*$^{Ndr/Ndr}$ and *Jag1*$^{+/+}$ biliary systems are similar in length.

entire tree to obtain comprehensive and accurate results. In summary, DUCT allowed analysis of curved and chord length measurements for the entire or defined regions of the injected trees. *Jag1$^{Ndr/Ndr}$* bile ducts recovered wild-type lengths postnatally, with a pronounced increase in peripheral tortuosity.

## Discussion

Precisely defining the three-dimensional (3D) architecture of healthy and diseased organs is a fundamental aspect of biology, and improved imaging methods would allow stricter characterization of animal models for human diseases. Until now, 3D liver analysis has been restricted by a lack of adequate tools and high auto-fluorescence of the tissue. Carbon ink injection is robust, but imaging is 2D, precluding 3D analysis of the architecture. In contrast, immunostaining and clearing allows 3D analysis, but success is variable and highly dependent on tissue fixation, antibody quality, penetrance and tissue autofluorescence. Here, we further advanced organ resin casting, which was previously used to analyze one system at a time (*Masyuk et al., 2003*; *Kline et al., 2011*; *Walter et al., 2012*). We developed a simple, robust and inexpensive method (DUCT) for simultaneous visualization and digitalization of two lumenized systems in mouse to analyze organ architecture. DUCT is completely independent of antibody staining, endogenous fluorescent proteins and is not sensitive to tissue fixation. Unlike whole mount immunohistochemistry techniques, DUCT provides information about the lumen, internal diameter, perfusion and connectivity of the injected tree. The most important limitation of DUCT is that it cannot visualize structures with a diameter under 5 µm, due to resin viscosity. We showed that resin casting, segmentation and 3D representation can be used as input for further investigation by visual qualitative assessment, and for in depth analysis by imaging softwares such as ImageJ or custom written MATLAB scripts. The pipeline for imaging and segmentation followed by detailed customized quantification of cellular and architectural mechanisms of two tubular networks could serve as a standard for whole organ analysis in animal models, and can be further adapted for a specific applications.

DUCT is based on radiopaque resin injection into multiple lumenized systems. First, identifying resins with sufficient contrast and low viscosity is crucial for scanning using computed tomography. Combining multiple resins with distinct contrasts can upscale the analysis to several networks simultaneously. In our study, it was imperative to use two fresh MICROFIL resins to obtain sufficiently distinct contrast to separate the two injected systems. Prolonged storage (~>3 months) of the MICROFIL resin leads to resin precipitation and a significant decrease in the resin radiopacity (for details see materials and methods). Future efforts to identify or develop differentially radiopaque substances would further accelerate the analysis pipeline. Second, in order to achieve a successful injection, an appropriate resin injection site for each system must be identified, and a suitable amount of pressure must be applied while avoiding bubble formation. Third, resin segmentation of the scanned sample is straightforward in well-injected, well-contrasted samples, but in samples with bubbles or poor contrast segmentation requires time-consuming manual correction and careful tracing of the resin throughout the whole organ to ensure a coherent network. Minor resin leakage can be digitally excluded during the image segmentation, but artefacts such as air bubbles in resin, non-homogenous resin contrast (in our set up caused by mixing of blue and yellow MICROFIL) and resin leakage due to lumen rupture (probably caused by high pressure during injection) slow down the analyses substantially (*Figure 7B*). Thus, a well-chosen and well-injected resin are a prerequisite for efficient downstream analyses.

Regarding instrumentation, the DUCT pipeline is not restricted to specific CT systems or acquisitions parameters, therefore samples can be imaged on any CT device with sufficient spatial resolution to study selected samples and their morphology. For further qualitative and quantitative analysis of the DUCT µCT data, robust computational power is necessary – and dedicated workstations with sufficient RAM memory (>64 MB RAM) are recommended, dependent on the volume of acquired data.

Our study describes a complex spatial adaptation of the biliary tree to postnatal BD paucity in a mouse model for Alagille syndrome (ALGS), with validation in samples from patients with ALGS. Based on our data and reported case studies, we propose a model in which *Jag1* mutant intrahepatic bile ducts (IHBDs) did not form during embryonic development (*Andersson et al., 2018*; *Alessandro et al., 2007*) but in some animals (and some patients, *Figure 2—figure supplement*

*1D*), bile ducts grew after birth (*Dahms et al., 1982*). The lumenized bile ducts formed from hilar to peripheral regions with different timings in the different liver lobes and in individual animals. The fully remodeled biliary system was tortuous, exhibited abrupt ends, and was hyper-branched in region 1 (summarized in *Figure 8*).

Our findings in the mice, using DUCT, may help to explain the poor prognostic value of biopsies in patients with ALGS. Our results suggest that the lack of predictive value between peripheral bile duct paucity, observed in diagnostic biopsies, and phenotype severity in patients with ALGS (*Mouzaki et al., 2016*) may reflect differences in bile duct growth and presence/absence in hilar versus peripheral regions (*Figure 1C*, *Figure 1—figure supplement 5B,C,E*). Mouzaki et al, showed that bile duct density was not predictive of outcome, instead, bilirubin levels, fibrosis, and cholestasis were correlated with disease presentation. In line with this, in the *Jag1^{Ndr/Ndr}* pups the total bilirubin levels correlated well with 3D postnatal bile duct growth, which was apparent from 3D resin-injected whole lobe analysis, but did not correlate with bile duct paucity in 2D sections of central

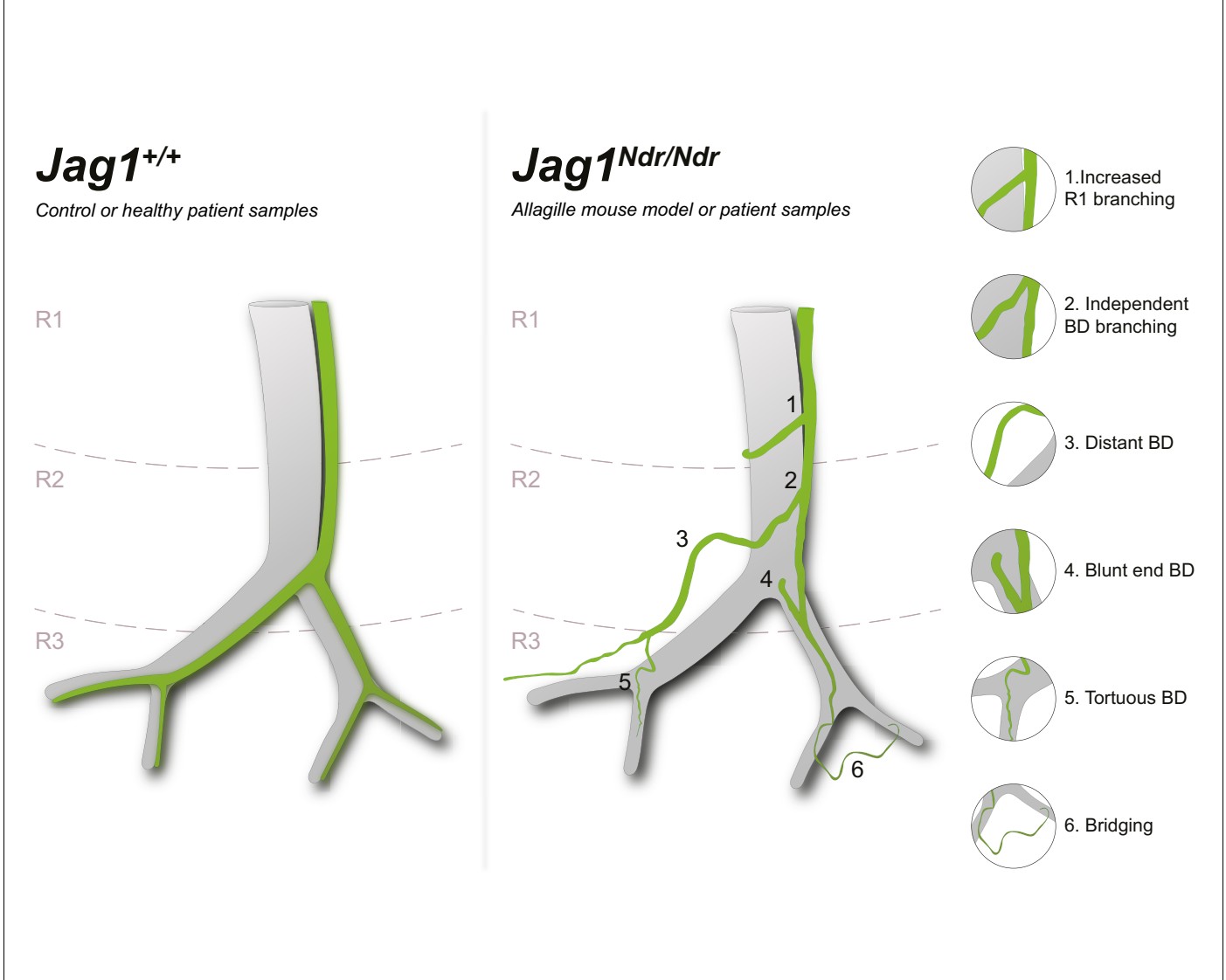

**Figure 7.** Schematic of *Jag1^{Ndr/Ndr}* biliary abnormalities in de novo generated bile ducts. Left panel depicts a simplified wild type or healthy human spatial arrangement of portal veins and bile ducts in three liver regions (R1, R2 and R3). Right panel illustrates a simplified adult *Jag1^{Ndr/Ndr}* regenerated biliary system displaying morphological abnormalities including (1) increased branching in region 1, (2) branching independent of the portal vein, (3) increased distance from portal vein, (4) abrupt/blunt endings facing the hilum, (5) peripheral tortuosity and (6) bridging between two portal veins. Independently branching, abruptly ending, parenchymal and tortuous bile ducts were confirmed in liver from patients with Alagille syndrome.

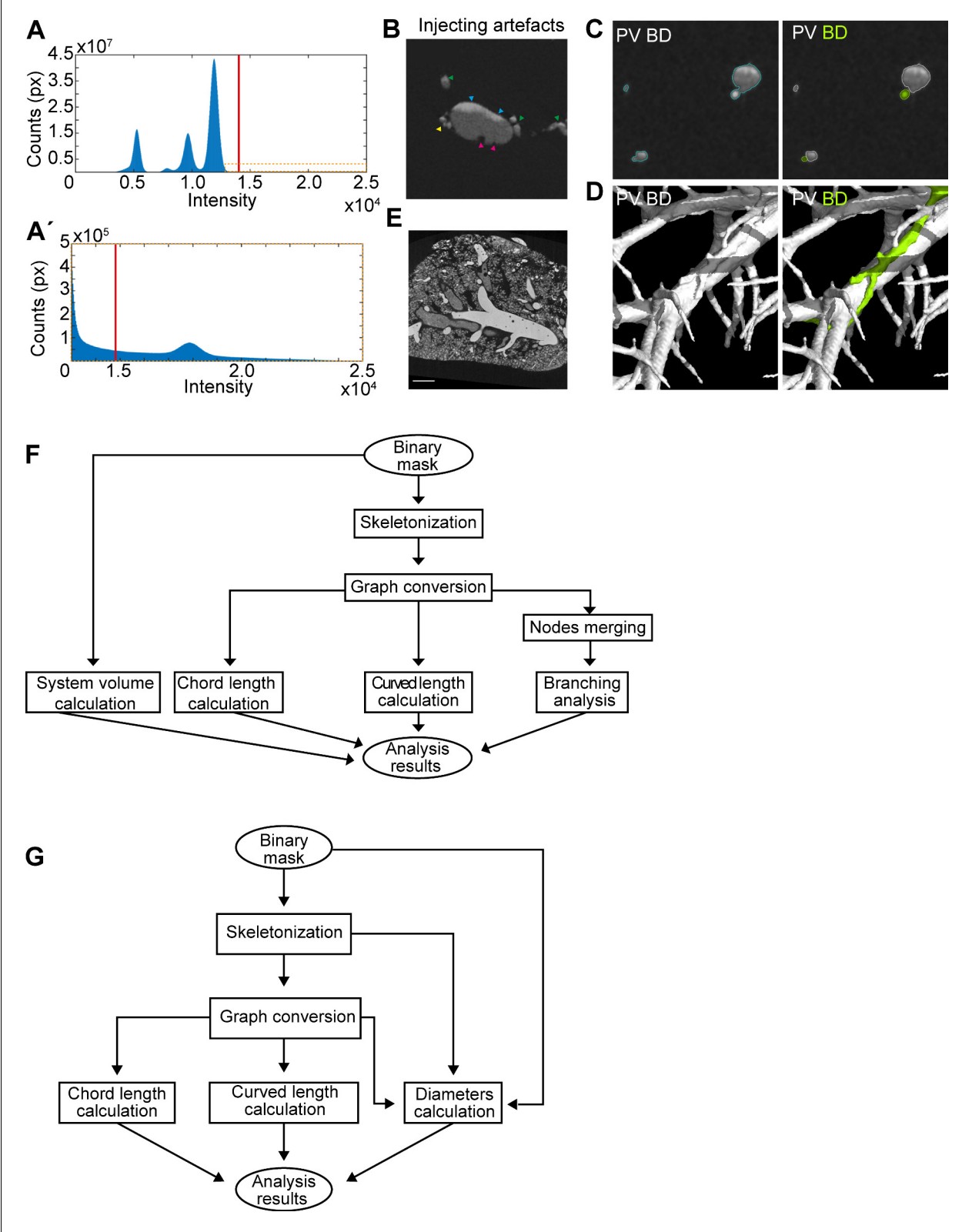

**Figure 8.** Micro computed tomography image processing. (**A**) µCT scan thresholding. Orange box in (**A**), magnified in (**A'**), shows MICROFIL intensity levels. (**B**) Resin injection artefacts including inadequate mixing of MICROFIL resin (blue arrowheads), bubbles in the resin which require manual correction (magenta arrowheads), and leakage due to excessive injection pressure or vessel/duct weakness (yellow arrowhead). Green arrowheads represent side branches. (**C**) Global thresholding separates old MICROFIL injected ducts and vessels from background tissue. BD and PV are identified
*Figure 8 continued on next page*

*Figure 8 continued*

manually. (**D**) 3D visualization of BD and PV after first segmentation (left panel) and after the systems separation. (**E**) Global thresholding separates fresh MICROFIL injected airways and vessels from each other and the background tissue. (**F**) Quantification pipeline schematic for the whole lobe (**G**) or only the main branch analysis.

liver. A 40% regrowth of lumenized bile ducts in *Jag1^{Ndr/Ndr}* pup was sufficient to reduce cholestatic burden to almost wild type levels, while not leading to a normal density of bile ducts in the periphery - emphasizing the importance of whole-liver architecture analyses.

Identifying and quantifying architectural defects such as branch length differences, stochastic branching, tortuosity, differences in portal-biliary distance and blunt end bile ducts is very challenging in sections (compare histology and 3D imaging in *Figures 2–6*). We demonstrated here that DUCT is a powerful method for visualization and semi-automated quantitative analysis of two lumenized biological systems in vivo. DUCT could be applied to other tubular networks including blood vessels and bronchi in lung (*Figure 1—figure supplement 1E*) or blood vessels and urinary ducts in kidney (*Wagner et al., 2011*; *Wei et al., 2006*). DUCT has multiple advantages over ink injections and iDISCO+, as 3D imaging with μCT avoids the drawbacks of tissue autofluorescence or poor antibody penetration. By injecting two resins into a single animal, it is possible to study the relation between these biological systems, which has not been previously reported. While experts in the field are careful to discriminate hilar and peripheral regions of the liver, carefully tracing organ structures for hundreds of micrometers in tissue sections is not standard practice and is a demanding endeavor. DUCT would be a suitable readout for testing drug compounds in mouse models for liver cholestatic diseases. With DUCT, it is now possible to map and quantify architecture of two networks in mouse models, setting the stage for an in depth understanding of how systems interact in health, disease, and regenerative processes.

# Materials and methods

## Key resources table

| Reagent type (species) or resource | Designation | Source or reference | Identifiers | Additional information |
|---|---|---|---|---|
| Antibody | Anti-Cytokeratin 7 (rabbit monoclonal) | Abcam | Cat # ab181598 RRID:AB_2783822 | iDISCO+ (1:1000) |
| Antibody | Anti-Cytokeratin 7 (mouse monoclonal) | Invitrogen/ThermoFisher Scientific | Cat # MA5-11986, clone: OV-TL 12/30 RRID:AB_10989596 | IHC (1:200) |
| Antibody | Anti-Cytokeratin 7 (mouse monoclonal) | Sigma - Aldrich | Cat # C6198 RRID:AB_476856 | iDISCO+ (1:2000) |
| Antibody | Anti-Cytokeratin 19 (rat monoclonal) | DSHB | Cat # TROMA-III RRID:AB_2133570 | IHC (1:50) |
| Antibody | Anti-human SOX9 (goat polyclonal) | RnD Systems | Cat # AF3075 RRID:AB_2194160 | IHC (1:100) |
| Commercial assay or kit | MICROFIL | Flow Tech Inc | Cat # MV120, MV-122 | DUCT |
| Software, algorithm | MATLAB | Mathworks | RRID:SCR_001622 | codes available: https://github.com/JakubSalplachta/DUCT |

## Experimental mice

All animal experiments were performed in accordance with Stockholm's Norra Djurförsöksetiska nämnd (Stockholm animal research ethics board, ethics approval numbers: N150/14, N61/16, N5253/19, N2987/20) regulations. Animals were maintained with standard day/night cycles, provided with food and water ad libitum, and were housed in cages with enrichment. For postnatal day 15 (P15) experiments, 10 wild type (*Jag1^{+/+}*) (eight males and two females) and 10 *Jagged1* Nodder (*Jag1^{Ndr/Ndr}*) littermate pups (five males and five females) were used for serum analysis. Within this group, nine *Jag1^{+/+}* and seven *Jag1^{Ndr/Ndr}* mice were injected with resin. All 16 animals were analyzed in 3D, revealing extensive heterogeneity that would necessitate performing DUCT on a large

number of animals to obtain significant quantitative data, while the bile duct paucity was obvious. 1 *Jag1$^{+/+}$* and 1 *Jag1$^{Ndr/Ndr}$* pair was therefore scanned and rendered in 3D. From this group, four *Jag1$^{+/+}$* and four *Jag1$^{Ndr/Ndr}$* left medial lobes were used for 2D liver sections and staining.

Adult animals were between 4.5 and 6.5 months old. In total, 18 *Jag1$^{+/+}$* and 6 *Jag1$^{Ndr/Ndr}$* animals were injected with resin for µCT. Quality control of injections (*Figure 1—figure supplement 2*) was performed on all livers during method development until surgery and injection technique resulted in well-injected livers. Three *Jag1$^{+/+}$* and three *Jag1$^{Ndr/Ndr}$* animals were used for the DUCT quantifications in adulthood. For liver histology, two *Jag1$^{+/+}$* and three *Jag1$^{Ndr/Ndr}$* mice were used. For ink injections, nine *Jag1$^{+/+}$* mice were used (four males and five females) and for iDISCO+ four *Jag1$^{+/+}$* and four *Jag1$^{Ndr/Ndr}$* mice were used (six males and two females). For lung 3D resin casting five *Jag1$^{+/+}$* mice were used (two males and three females). Samples were not blinded for investigation since the phenotype is overt and the genotype is therefore obvious to the experimenter. The animals were maintained on a mixed C57bl6J/C3HeN background. *Jag1$^{Ndr/+}$* (Nodder) mice were bred and genotyped as previously described (*Andersson et al., 2018*).

## Patient samples

Collection of liver samples and clinical data from patients or donors was approved by the Swedish Ethical Review Authority (2017/269-31, 2017/1394–31). Samples from patients with severe Alagille syndrome (four) were obtained at time of liver transplant from extirpated liver. Samples were obtained with a consent to be used for research according to ethical permit 2017/269–31. Samples were dissociated for primary cell culture e.g. organoids (data not shown), and a matching sample was formalin fixed or fresh-frozen for comparative analyses. Liver tissue samples from patients with mild Alagille syndrome (six) were obtained for clinical follow-up purposes and were retrospectively analyzed. The liver material was obtained within the framework of clinical patient care can be analyzed retrospectively without the need for consent according to ethical permit 2017/1394–31. Healthy controls (two) were left-over donor material, or from organ donation post-mortem. The liver function tests were obtained during routine biochemical analyses.

## MICROFIL injections

MICROFIL (Flow Tech Inc) was prepared as follows. Yellow MICROFIL (Y) cat. #MV-122 was diluted with clear MICROFIL (C) cat. # MV-Diluent in 3:1 (Y:C). Blue MICROFIL (B) cat. #MV-120 was diluted 1:1 (B:C) with clear MICROFIL. Diluted yellow MICROFIL was mixed with diluted blue MICROFIL 1:1 creating a green MICROFIL. Yellow MICROFIL was injected into common bile duct (CBD) or pulmonary artery (PA). Green MICROFIL was injected into portal vein (PV) or trachea (TA). 1 ml of diluted MICROFIL is mixed with 50 µl of hardener (supplied by Flow Tech Inc) prior injection.

Postnatal day 15 (P15) mice were sacrificed by decapitation and perfused through the heart with 3 ml of Hanks' Balanced Salt solution (HBSS) (Life Technologies cat. # 14025092). Adult mice were sacrificed by $CO_2$ inhalation and perfused through the heart with HBSS for 3 min (perfusion rate 5 ml / 1 min). For liver resin injections, the mice were perfused trough the left ventricle, for lung resin injections the mice were perfused though the right ventricle.

### Injection into CBD

A small transversal incision was made in inferior vena cava with spring scissor to release the liver vascular pressure. CBD was exposed by moving aside the liver and intestine and cleaned from surrounding tissue in area about 5 mm long. Silk suture (Agnthos AB cat. #14757) was loosely wrapped around the cleaned CBD. A longitudinal CBD incision was made at the spot where CBD enters the pancreas next to sphincter of Oddi by spring scissor. The tubing (PE10, BD Biosciences cat. # 427401) ~15 cm long was prepared by stretching one side of the tube until the diameter becomes thin enough to fit into CBD. Diagonal cut is made at the tip of the tubing while the other side contains needle (27G) connected to the syringe filled with MICROFIL. The tubing connected to a syringe and filled with a yellow MICROFIL was inserted into the CBD, the suture around the CBD can be tighten to secure the tubing in place. Yellow MICROFIL was injected into the CBD until resistance was met or MICROFIL spots were visible on the liver surface. Massaging the liver with cotton swab while injecting helped to disperse the MICROFIL. The tubing was removed and silk suture was tightened around the CBD to prevent leakage.

## Injection into PV

PV was cleaned from surrounded tissue. A small incision was made in PV using spring scissor. Silk suture was loosely wrapped around the cleaned PV above the incision. Tubing (PE10) ~15 cm long connected to (27G) needle was inserted into the PV incision and secured with silk suture. Green MICROFIL was injected into the PV until blood vessels on the surface were filled or resistance was met. Massaging the liver with a cotton swab while injecting helped to introduce the MICROFIL. The tubing was removed and silk suture was tightened around the PV to prevent leakage.

Liver was dissected out and placed at 4°C overnight (ON) for MICROFIL to solidify. The next day the liver was fixed with 3.7% formaldehyde solution (FA) (Sigma-Aldrich cat. #F1635) diluted in Dulbecco's phosphate-buffered saline (DPBS) (Life Technologies cat. # 14190144). After 24 hr, liver was washed and kept in DPBS. Liver was separated into lobes. The left lateral lobe was placed in 50% methanol (Sigma-Aldrich, cat. # 322415) for 4 hr and into 100% methanol ON. Further, the lobe was placed in benzyl alcohol (Sigma-Aldrich, cat. #402834) and benzyl benzoate (Sigma-Aldrich, cat. #B6630) (BA:BB 1:2) solution until transparent. The right medial lobe (only FA fixed) was used for µCT scanning. Liver lobe images of right medial lobe (P15) and left lateral lobe (P15 and adult) were taken using a stereomicroscope Stemi 305 (Carl Zeiss Microscopy) with a PowerShot S3 IS camera (Canon) or iPhone6 connected to a LabCam adapter.

## Injection into lung

The mouse heart was pulled toward the liver to expose the pulmonary artery (PA) and pinned down through the heart apex with a 1 ml empty syringe connected with a needle. A silk suture was wrapped loosely around the PA as close to the heart as possible. A small incision was made in the right ventricle with spring scissors. Tubing (PE50, BD bioscience, cat #427411) ~15 cm long (stretched at the tip) connected to (23G) needle was inserted into the PA through the incision in the right ventricle and tightened with the suture. A total of 1 ml of DPBS was injected into the lung via PA to remove all the remaining blood. Afterwards, to expand the collapsed lung, the trachea was exposed and cleaned from surrounding tissue. A silk suture was loosely wrapped around the trachea and a small incision was made into the trachea with spring scissors. Tubing (PE50) ~15 cm long connected to (23G) needle was inserted into the trachea and tightened with the suture. One ml of DPBS was injected into the lungs via trachea – this inflates the collapsed lungs. A 1 ml syringe filled with yellow MICROFIL was connected to the tubing inserted into the PA, and MICROFIL was injected into the PA vasculature until all the blood vessels were filled. Massaging the lung with a cotton swab while injecting helped to disperse the MICROFIL. After the vasculature was completely filled, the tubing was removed and the suture around the PA tightened to prevent MICROFIL leakage. For airways injection, a 1 ml syringe filled with green MICROFIL was connected to the tubing inserted into the trachea, and MICROFIL was injected into the trachea until the lung was entirely filled with MICROFIL. Massaging the lung with cotton swab while injecting again helped to disperse the MICROFIL. After the lung was completely filled, the tubing was removed and the suture around the trachea tightened to prevent MICROFIL leakage.

Lungs were dissected out and placed at 4°C ON to allow the MICROFIL to solidify. The next day the lung was fixed with 3.7% FA diluted in DPBS. After 24 hr, lungs were washed and kept in DPBS. Lungs were separated into lobes and the right superior lobe was used for µCT scanning.

## Ink injections

Mice were sacrificed by $CO_2$ inhalation and transcardially perfused with HBSS for 3 min (perfusion rate 5 ml/1 min).

Injection into CBD and PV. CBD and PV were accessed in the same way as described for MICROFIL injections. When injecting with ink, there is no need to tighten the CBD with silk suture as the ink is not leaking out. Black ink (Higgins cat. #44032) was injected into the CBD until BDs on the surface were filled or resistance was met. White ink (Higgins cat. #44032) was injected using PE50 tubing into the PV until blood vessels on the surface were filled or resistance was met. Liver was dissected out and separated into lobes. All lobes were cleared in BABB as described above. Liver ink images of right medial lobe were taken under stereomicroscope Stemi 305 (Carl Zeiss Microscopy) using PowerShot S3 IS camera (Canon).

## Whole mount immunohistochemistry

Mice were anesthetized by isoflurane inhalation (~2%) and transcardially perfused with HBSS for 3 min (perfusion rate 5 ml/1 min) and 10% neutral buffered formalin (NBF) for 5 min. Liver was dissected out and further immersion fixed with 10% NBF ON at 4°C. The next day liver was washed and kept in DPBS and separated into lobes. Right medial lobe was stained and cleared following the iDISCO+ protocol and imaged by light sheet microscope by Gubra (Denmark).

Fixed and washed samples were dehydrated in methanol/$H_2O$ gradient: 20%, 40%, 60%, 80% and 2 × 100% methanol, each step 1 hr at room temperature (RT). The samples were bleached in cooled fresh 5% $H_2O_2$ in methanol ON at 4°C. The samples were subsequently rehydrated in methanol/PBS series: 80%, 60%, 40%, 20%, with 0.2% Triton X-100, 1 hr each at RT. They were washed in PBS with 0.2% Triton X-100 (PTx.2) for 2 × 1 hr at RT.

## Whole organ immunolabeling (iDISCO+)

Samples were incubated in permeabilization solution at 37°C for 3 days. Blocking is carried out in blocking solution at 37°C for 2 days. The samples were incubated with primary antibody in PTwH/5% DMSO/3% donkey serum at 37°C for 7 days. They were washed in PTwH for 1 × 10 min, 1 × 20 min, 1 × 30 min, 1 × 1 hr, 1 × 2 hr and 1 × 2 days. Samples were incubated with secondary antibody in PTwH/3% donkey serum at 37°C for 7 days, followed by washes in PTwH: 1 × 10 min, 1 × 20 min, 1 × 30 min, 1 × 1 hr, 1 × 2 hr and 1 × 3 days. All steps were performed in tightly closed tubes to minimize evaporation and oxidation.

### Solutions for iDISCO

PTx.2 (1L): 100 ml PBS 10x, 2 ml TritonX-100
PTwH (1L): 100 PBS 10x, 2 ml Tween-20, 1 ml of 10 mg/ml heparin stock solution
Permeabilization solution (500 ml): 400 ml PTx.2, 11.5 g glycine, 100 ml DMSO
Blocking solution (50 ml): 42 ml PTx.2, 3 ml donkey serum, 5 ml DMSO
Secondary antibody: Alexa Fluor 488 (dilution 1:1000, Life technologies)

### Tissue clearing

Tissue was cleared in methanol/$H_2O$ series: 20%, 40%, 60%, 80%, and 100% for 1 hr each at RT. Samples were incubated for 3 hr (with shaking) in 66%DCM (Dichloromethane)/33% methanol at RT and in 100% DCM 15 min 2x (with shaking) to remove traces of methanol. They were incubated in DiBenzyl Ether (DBE) (without shaking).

### Light sheet microscopy

Tissue samples were imaged using a light sheet microscope (UltramicroscopeII, Miltenyi). DBE was used as clearing agent during data acquisition. Data was collected at room temperature using Lavision ultramicroscope system and MV PLAPO 2X C/0.5 objective with dry lens, RI correction collar using Andor Zyla 4.2 Plus sCMOS camera. CK7 staining was detected with AF790 (Alexa Fluor, Life Technologies). The acquisition software used was ImSpector (LaVision biotech).

## Liver immunohistochemistry

5 μm FFPE-liver (mouse and human) sections were deparafinized and rehydrated through consecutive baths of xylene (cat. #28975.325, VWR) and isopropanol (cat. #K50655934838, Merck). Endogenous peroxidase was blocked by immersion of the slides in methanol (cat. #322415, Sigma-Aldrich) containing 0,3% $H_2O_2$ (cat. #H1009, Sigma-Aldrich) for 15 min and rehydration was finalized by rinsing the slides in tap water. Heat-induced epitope retrieval was done using citrate buffer (PH 6.0) for 20 min in a pressure cooker. After blocking of the sections with 2% BSA (cat. #A7906, Sigma-Aldrich) for 20 min, slides were incubated for 1 hr at 37°C with primary antibody (antibodies used are listed in Key resources table). Anti-mouse (cat. #G21040, dilution: 1/1000, Invitrogen) or anti-goat (Impress, cat. #MP7405, Vector), respectively, HRP-coupled secondary antibody was applied for 30 min at 37°C and revealed with DAB for 30 s (cat. #K3468, Dako). After counterstaining with hematoxylin (cat. #HX86014349, diluted 1/5, Merck), the sections were dehydrated in consecutive baths of ethanol (cat. #20821.310, VWR), isopropanol (cat. #K50655934838, Merck), and xylene

(cat. #28975.325, VWR) to finally be mounted with hardening medium (Eukitt, cat. #03989, Sigma-Aldrich).

## Liver section image acquisition

Chromogenic stained images were taken with AxioImager (Carl Zeiss) microscope, Axiocam 503 color camera using Plan-Apochromat 5x/0.16, Plan-Apochromat 10x/0.45 M27, Plan-Apochromat 20x/0.8 M27 and Plan-Apochromat 40x/1.4 Oil DIC (UV) VIS-IR M27 objectives at room temperature. The acquisition software used was Zen Blue (Carl Zeiss).

## Image processing

Whole mount liver images cleared with iDISCO+ were initially processed in ImageJ for maximum z-projection and segmentation. The images were filtered using the unsharp mask and integral image filter function. Images were next processed in Amira. In Amira images were filtered using the Gaussian filter and background detection correlation. Images were manually segmented. The manual segmentation was further traced using the autoskeleton function. The skeletons were further analyzed in Amira for length, volume and branching.

Images of ink injected liver were proceeds for filament tracing. Bile duct and portal vein filament tracing was performed using Amira. The images were filtered using the unsharp mask and mean filter. The signal was manually segmented to remove artificial signal. The manual segmentation was further traced using the autoskeleton function. The skeletons were analyzed in Amira for length, volume and branching. For double ink injection (*Figure 1—figure supplement 1A*) the background was changed for esthetic purposes using the lasso tool in Adobe Photoshop.

DUCT 2D slices were exported from MyVGL (Volumegraphics) and processed in ImageJ for maximum contrast and brightness.

## Liver sections IHC were processed in ImageJ for contrast and brightness

P15 MICROFIL injected and BABB-cleared left lateral and right medial lobe were analyzed in ImageJ. The total liver area was measured followed by the measurement of the liver area covered by MICROFIL injected bile ducts. The percentage of liver containing bile ducts was calculated.

## Blood serum collection and analysis

Blood from P15 pups was collected from the trunk after decapitation into 1.5 ml tubes. The serum was allowed to clot at room temperature. The blood was centrifuged for 15 min at 17,000 g at room temperature. The serum was stored at −80°C until analyzed. Serum was sent to the Swedish University of Agricultural Sciences for analysis of alanine aminotransferase (ALT), alkaline phosphatase (ALP), aspartate aminotransferase (AST), albumin (Alb), and total bilirubin.

## MicroCT measurement

The system GE Phoenix v|tome|x L 240 (GE Sensing and Inspection Technologies GmbH, Germany) equipped with nanofocus X-ray tube (180 kV/15 W) was used for the tomographic measurements that were carried out in the air-conditioned cabinet (fixed temperature 21°C). The samples were adapted for this temperature before the measurement to prevent any thermal expansion effect. To prevent any sample motion during the scanning, the samples were placed in 15 ml Falcon tube, filled with 1% agarose gel. The tomographic reconstruction of acquired data was performed using GE phoenix datos|x 2.0 software. The voxel resolution was fixed for all the adult liver samples at 12 μm, except one (sample #2401, 8 μm). For all the P15 liver samples the voxel resolution was fixed at 6.5 μm and for the lung lobe sample at 8 μm. Detailed overview of used acquisition parameters is stated in *Table 2*.

## MicroCT data segmentation

The identification and segmentation of both tubular systems (e.g. bile duct (BD) and portal vein (PV) for liver samples and pulmonary artery and airways for lung samples) in each CT cross-section was necessary for further analysis of each system. The segmentation was based on differential contrast between the resin and the soft tissue. Two different resins were used to identify the individual

tubular systems. The differential contrast for system identification was highly dependent on the freshness of the MICROFIL.

The resin segmentation was performed by global thresholding in VG Studio MAX 3.3 (Volume Graphics GmbH, Germany) software together with manual corrections where necessary. The threshold value was determined based on the histogram shape and visual evaluation of a selected cross-section (*Figure 8A,A'*). The resin cast, especially when using an old MICROFIL, could contain artefacts caused by poor contrast, insufficient filling or high injection pressure. These artefacts include air bubbles in resin, non-homogenous resin contrast (caused by mixing of blue and yellow MICROFIL) and resin leakage due to lumen rupture (probably caused by high pressure during injection), (*Figure 8B*). Therefore, the thresholding step was supplemented with manual corrections to create smooth, continuous and solid canal masks. Furthermore, the cut-off for the smallest distal canal included in the mask is considered an area of at least four voxels.

Next, the individual tubular systems were identified in the segmented resin mask (*Figure 1—figure supplement 12D and G*). When fresh MICROFIL was used, it was possible to identify each system by global thresholding with threshold value determined based on histogram shape and visual evaluation of CT data. However, in case of the old MICROFIL (used for adult liver samples) the resin absorption properties were not distinguishable in CT data. In most regions, both tubular systems blended with each other in one continuous region (*Figure 8C, D* left panel). Manual segmentation was therefore necessary to ensure the correct identification of both systems (*Figure 8C, D* right panel). The manual segmentation was performed by outlining the BD regions in every slice of the CT data. VG Studio automatically creates 3D render based on the regions outlined in CT sections.

## MicroCT data analysis

The adult liver was analyzed using a custom-written algorithm and freely available Matlab codes (Version R2017a, The MathWorks Inc, Natick, MA). The algorithm was designed to analyze morphological parameters of the BD and PV systems, and is compatible with the 3D binary masks. Two separate masks of BD and PV system were generated and the analysis was divided in two independent parts. First, the analysis of the entire portal vein and biliary system and second, analysis of the corresponding main branch (=the longest branch) of each system (*Figure 6—figure supplement 1A*). For detailed analysis and comparison of the whole system versus only the main branch, two algorithms, described by the diagrams in *Figure 8F, G*, were developed. They differ in the input data and the evaluated parameters. For both algorithms, the first step is to create a 3D skeleton of the input binary mask.

## Skeletonization of binary masks

The 3D skeleton was derived using the homotopic thinning algorithm described in *Lee et al., 1994* specifically optimized for Matlab implementation by Kollmannsberger (*Kerschnitzki et al., 2013*), (online source: https://www.mathworks.com/matlabcentral/fileexchange/43400-skeleton3d).

**Table 2.** Settings parameters of the GE Phoenix v|tome|x L 240 system.

| Sample | Voxel size | Acceleration voltage | X-ray tube current | Exposition time | Number of projections |
|---|---|---|---|---|---|
| 2401 | 8 μm | 80 kV | 160 μA | 600 ms* | 2500* |
| 2404 | 12 μm | 80 kV | 160 μA | 600 ms* | 2500* |
| 2405 | 12 μm | 80 kV | 160 μA | 600 ms* | 2500* |
| 2431 | 12 μm | 80 kV | 160 μA | 600 ms* | 2500* |
| 2713 | 12 μm | 80 kV | 160 μA | 334 ms[†] | 1900[†] |
| 2714 | 12 μm | 80 kV | 160 μA | 334 ms[†] | 1900[†] |
| N864 | 6.5 μm | 80 kV | 160 μA | 400 ms[†] | 1800[†] |
| N865 | 6.5 μm | 80 kV | 160 μA | 400 ms[†] | 1800[†] |
| *Jag1*[+/+] lung | 8 μm | 80 kV | 160 μA | 400 ms[†] | 2000[†] |

*Flat panel DXR250 (2048 px ×2048 px, pixel size 200 μm).
[†]Flat panel dynamic 41|100 (4048 px ×4048 px, pixel size 100 μm with binning 2).

Calculated 3D medial axis skeleton was subsequently converted to a network graph, using algorithm described in *Kerschnitzki et al., 2013* (online source: https://www.mathworks.com/matlabcentral/fileexchange/43527-skel2graph-3d). Resulting network graph is formed by nodes and links between them (*Figure 1—figure supplement 12D and G*).

## Liver region subdivision

Liver was separated into three regions: R1 (approximately hilum), R2 (approx. intermediate) and R3 (approx. periphery). The optimal region size was calculated as a summary of average PV branch lengths per generation and divided by three (detailed branching generation subdivision *Table 3* for PV and *Table 4* for BD). Subsequently the branching generations were assigned to a region by matching the summary of branch length per generation to optimal region size. The same region was applied for both PV and BD analysis with the exception of one $Jag1^{Ndr/Ndr}$ sample (#1 or 2714) where the bile duct optimal region size was greater than portal vein.

Regions were assigned in order that total average lengths of each generation within each region yielded an equal size of R1, R2, and R3. Each sub-column represents one animal.

The distribution of BD branches within each region were quantified based on regions defined by the PV system (*Table 4*). Each sub-column represents one animal.

## Branching analysis

Branching points analysis was programmed in Matlab to analyze the distance between BD and PV branching point. This parameter was calculated using 3D Euclidean distances between the BD branching points and the nearest branching point from PV system. The data is represented as cumulative sum of percentage of BD branching point at a given distances between BD and PV branching points (from 0.015 mm to 3 mm).

For branch length analysis the structure of BD and PV trees were first reconstructed in 3D, using the Analyze Skeleton toolbox in ImageJ, which provided the three-dimensional coordinates of all branch points for both BD and PV, as well as the connectivity of the graph. Next, we computed for each branch the length along its path to the Euclidean distance between its extremities (branch points). To calculate the generation number of branches (both for the PV and BD structures), we manually defined the origin of the ducts and vessels as generation 1, and computed generation number as the number of generation branches separating a given branch from the origin. To distinguish side branching events, we calculated the angle between a branch and its 'parent' by computing the dot product p of both their unit vectors. A branch with p>0.95 with its parent branch was considered to belong to the same generation. We then computed distributions of length for the BD and PV structures as a function of generation number. Each generation was assigned to a region R1, R2, or R3.

The branch distribution in each region was calculated as a summary of number of branches per generation in a given region. The sum of branch numbers of each region is displayed as a proportion of the total number of branches per sample.

The number of bi-furcations (i.e. one input and two outputs), tri-furcations (i.e. one input and three outputs) and quadri- and more-furcations (one input and more than three outputs), were assessed in Matlab based on binary mask skeleton nodes that were divided into endpoints and branching points. Branching points closer than 0.2 mm (this threshold value was derived based on visual assessment and knowledge of the system) were merged together and further represented by one node.

**Table 3.** PV branching generation distribution into liver regions.

| PV generation # | $Jag1^{+/+}$ (3 animals) | | | $Jag1^{Ndr/Ndr}$ (3 animals) | | |
|---|---|---|---|---|---|---|
| R1 | 1–4 | 1–3 | 1–4 | 1–4 | 1–3 | 1–3 |
| R2 | 5–10 | 4–9 | 5–10 | 5–8 | 4–7 | 4–7 |
| R3 | 11–18 | 10–14 | 11–17 | 9–15 | 8–12 | 8–11 |

**Table 4.** BD branching generation distribution into liver regions.

| BD generation # | Jag1+/+ (3 animals) | | | Jag1Ndr/Ndr (3 animals) | | |
|---|---|---|---|---|---|---|
| R1 | 1–4 | 1–4 | 1–2 | 1–4 | 1–6 | 1–6 |
| R2 | 5–7 | 5–7 | 3–5 | 5–9 | 7–10 | 7–10 |
| R3 | 8–12 | 8–11 | 6–7 | 10–16 | 11 | 11–12 |

## Gap analysis between bile duct and portal vein

To evaluate the gap between BD and PV the surface distances were calculated in Matlab for each BD skeleton point by detecting the nearest PV skeleton point and connecting the two points with a line and measuring the non-resin area on this line (zero area in the input binary masks). Surface distance was then calculated using 3D Euclidean distance between the detected non-resin voxel coordinates. The data is represented as cumulative sum of percentage of BD at a given distance from PV (from 0.015 mm to 1.5 mm). The maximum distance between BD and PV for each liver sample was depicted in a separate graph.

## Tortuosity measurements

To quantify length and tortuosity, total (curved) and theoretical (chord) lengths were measured in Matlab for the whole system length and for the corresponding main branch. The curved length was defined as a cumulative sum of 3D Euclidean distances between neighboring graph points (i.e. links forming points) multiplied by voxel size. The chord length was defined as cumulative sum of 3D Euclidean distances between neighboring nodes multiplied by voxel size. The chord length therefore reflects system length where any nodes are connected by links with the shortest possible length. To analyze the relationship between BD and PV a length of the BD was divided by a PV length (curved or chord). Tortuosity was calculated as curved length divided by chord length of the same system and distributed into regions based on the generation number as previously described. Tortuosity was assessed in %, as BD and PV are not straight lines the actual tortuosity measurements were subtracted by 100% (perfectly straight line).

## Volume analysis

Total system volume was calculated in Matlab by multiplying a number of voxels representing PV or BD by volume of one voxel. The relationship between BD and PV volumes was addressed by dividing the BD volume by PV volume.

## Diameter measurements

The main branch diameter was calculated in Matlab every 1.5 mm along the total length of the main branch. The radius was defined as the minimal distance from the skeleton to the segmented area boundary in the input binary mask (i.e. border between background and area of interest). This boundary was calculated using a two-step procedure. In the first step, the input map was eroded using a 3D spherical shaped structural element with one pixel radius. Subsequently, the eroded area was subtracted from the original binary mask. This resulted in a binary mask representing the boundary between the background and the area of interest. One radius value at a given skeleton point was then expressed as the minimum distance from that point to the mask boundary. This was calculated, using the minimal value search in the intersection of the boundary mask and the distance map from that point. The distance map from a given skeleton point was calculated as 3D Euclidean distance of the spatial coordinates. Subsequently the diameter value was calculated as the minimum distance to the boundary area multiplied by 2. To avoid any misrepresentation, the one final diameter value at a given point (every 1.5 mm of branch length) was calculated as a mean value of a diameter at that point and diameters at four neighboring points (two on each side). PV and BD diameters were divided into three areas: hilum, intermediate and periphery. Hilar region represents distance from 0 to 1.5 (sample #2401) or 0–3 mm (other samples), Intermediate region is from 3 to 6 mm (sample #2401) or 4.5 mm – 9 mm (other samples), Periphery is from 7.5 mm – 9 mm (sample #2401) or 10.5–13.5 mm (other samples). BD to PV diameter ratio was calculated by dividing BD diameter at a given region by PV dimeter of the same region.

## Statistical analysis

$Jag1^{+/+}$ and $Jag1^{Ndr/Ndr}$ data were tested for significant differences using multiple tests based on the type of experiment and data distribution. Student´s t-test (*Figures 1D, G* and *3D*, *Figure 1—figure supplement 12H*, *Figure 6—figure supplement 1B–E, H–K*). Kolmogorov-Smirnov test (on raw data, graph depicts cumulative sum) (*Figures 3C* and *4C*). Mann-Whitney test (*Figure 4—figure supplement 1*). Wilcoxon test (*Figure 1—figure supplement 5D*). Two-way ANOVA (*Figures 1H* and *5B–D*, *6E*, *Figure 1—figure supplements 5C* and *12I*, *Figure 6—figure supplement 1F and G*) followed by Sidak's multiple comparisons test. Spearman correlation (*Figure 1E*). A p value below 0.05 was considered statistically significant. The statistical analysis was done in Prism 9 (GraphPad).

## Acknowledgements

Grant support: ERA: Work in ERA lab is supported by the Swedish Research Council, the Center of Innovative Medicine (CIMED) Grant, Karolinska Institutet, and the Heart and Lung Foundation, and the Daniel Alagille Award from the European Association for the Study of the Liver. One project in ERA lab is funded by ModeRNA, unrelated to this project. The funders have no role in the design or interpretation of the work. SH has been supported by a KI-MU PhD student program, and by a Wera Ekström Foundation Scholarship. We are grateful for support from Tornspiran foundation to NVH. JK: This research was carried out under the project CEITEC 2020 (LQ1601) with financial support from the Ministry of Education, Youth and Sports of the Czech Republic under the National Sustainability Programme II and CzechNanoLab Research Infrastructure supported by MEYS CR (LM2018110) . UL: The financial support from the Swedish Research Council and ICMC (Integrated CardioMetabolic Center) is acknowledged. JJ: The work was supported by the Grant Agency of Masaryk University (project no. MUNI/A/1565/2018). We thank Kari Huppert and Stacey Huppert for their expertise and help regarding bile duct cannulation and their laboratory hospitality. We also thank Nadja Schultz and Charlotte L Mattsson for their help with common bile duct cannulation. We thank Daniel Holl for his help with trachea cannulation. We thank Nikos Papadogiannakis for his assistance with mild Alagille biopsy samples and discussion. We thank Karolinska Biomedicum Imaging Core, especially Shigeaki Kanatani for his help with image analysis. We thank Jan Masek and Carolina Gutierrez for their scientific input in manuscript writing. We thank Peter Ranefall and the BioImage Informatics (SciLife national facility) for their help writing parts of the MATLAB pipeline. The TROMA-III antibody developed by Rolf Kemler was obtained from the Developmental Studies Hybridoma (DSHB) Bank developed under the auspices of NICHD and maintained by The University of Iowa, Department of Biological Sciences, Iowa City, IA52242. We thank Goncalo M Brito for all illustrations. This work was supported by the European Union (European Research Council Starting grant 851288 to E.H.).

## Additional information

### Funding

| Funder | Grant reference number | Author |
|---|---|---|
| Karolinska Institutet | 2-560/2015-280 | Emma Rachel Andersson |
| Stockholms Läns Landsting | CIMED (2-538/2014-29) | Emma Rachel Andersson |
| Ragnar Söderbergs stiftelse | Swedish Foundations' Starting Grant | Emma Rachel Andersson |
| European Association for the Study of the Liver | Daniel Alagille Award | Emma Rachel Andersson |
| Swedish Heart-Lung Foundation | 20170723 | Emma Rachel Andersson |
| Vetenskapsrådet | 2019-01350 | Emma Rachel Andersson |
| Ministerstvo Školství, Mládeže a Tělovýchovy | LQ1601 | Jozef Kaiser |
| Ministerstvo Školství, Mládeže | LM2018110 | Jozef Kaiser |

a Tělovýchovy

| European Research Council | Starting grant 851288 | Edouard Hannezo |

The funders had no role in study design, data collection and interpretation, or the decision to submit the work for publication.

## Author contributions

Simona Hankeova, Conceptualization, Data curation, Formal analysis, Validation, Investigation, Visualization, Methodology, Writing - original draft; Jakub Salplachta, Conceptualization, Software, Formal analysis, Investigation, Visualization, Methodology, Writing - original draft; Tomas Zikmund, Conceptualization, Software, Formal analysis, Supervision, Investigation, Visualization, Methodology, Writing - review and editing; Michaela Kavkova, Formal analysis, Investigation, Visualization, Methodology, Writing - review and editing; Noémi Van Hul, Formal analysis, Validation, Investigation, Visualization, Writing - original draft; Adam Brinek, Jakub Laznovsky, Software, Visualization, Writing - review and editing; Veronika Smekalova, Visualization, Writing - review and editing; Feven Dawit, Investigation; Josef Jaros, Validation, Writing - review and editing; Vítězslav Bryja, Urban Lendahl, Funding acquisition, Writing - review and editing; Ewa Ellis, Resources, Writing - review and editing; Antal Nemeth, Björn Fischler, Data curation; Edouard Hannezo, Conceptualization, Investigation, Writing - review and editing; Jozef Kaiser, Conceptualization, Resources, Funding acquisition, Project administration, Writing - review and editing; Emma Rachel Andersson, Conceptualization, Resources, Data curation, Formal analysis, Supervision, Funding acquisition, Methodology, Writing - original draft, Project administration

## Author ORCIDs

Simona Hankeova https://orcid.org/0000-0002-7797-3818
Jakub Salplachta https://orcid.org/0000-0002-0149-7843
Noémi Van Hul https://orcid.org/0000-0003-1410-8808
Vítězslav Bryja http://orcid.org/0000-0002-9136-5085
Ewa Ellis http://orcid.org/0000-0002-3057-5337
Edouard Hannezo http://orcid.org/0000-0001-6005-1561
Emma Rachel Andersson https://orcid.org/0000-0002-8608-625X

## Ethics

Human subjects: Collection of liver samples from patients or donors was approved by the Swedish Ethical Review Authority (2017/269-31, 2017/1394-31). Samples were obtained with a consent to be used for research according to ethical permit 2017/269-31.

Animal experimentation: All animal experiments were performed in accordance with Stockholm's Norra Djurförsöksetiska nämnd (Stockholm animal research ethics board, ethics approval numbers: N150/14, N61/16, N5253/19, N2987/20) regulations.

## Decision letter and Author response

Decision letter https://doi.org/10.7554/eLife.60916.sa1
Author response https://doi.org/10.7554/eLife.60916.sa2

# Additional files

## Supplementary files

- Transparent reporting form

## Data availability

Our MATLAB pipeline is deposited in Github: https://github.com/JakubSalplachta/DUCT. Copy archived at https://archive.softwareheritage.org/swh:1:rev:6b0b0eb88bbaf9bfc4f8ee42ca-fa4c122866fbba/. All data generated or analysed during this study are included in the manuscript and supporting files. Source data files have been provided for Figures 3 and 4.

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
