## [Decision Letter]

**Acceptance summary:**

You have developed and applied an exciting approach to map development of the biliary tree and portal vasculature and thereby, clarify the pathogenesis and progression of Alagille syndrome in mouse models of that disease. The new data from Alagille syndrome patients and new studies that you performed in the mouse model at earlier stages of the disease provide compelling support for your initial conclusions. Further, the added work that you did to assess branching morphogenesis in the lung strengthens your initial contention that this new technical approach provides a broadly relevant tool for researchers in developmental biology and regenerative medicine.

**Decision letter after peer review:**

Thank you for submitting your article "DUCT reveals architectural mechanisms contributing to bile duct recovery in a mouse model for Alagille syndrome" for consideration by *eLife*. Your article has been reviewed by three peer reviewers, including Anna Mae Diehl as the Reviewing Editor and Reviewer #1, and the evaluation has been overseen by Didier Stainier as the Senior Editor.

The reviewers have discussed the reviews with one another and the Reviewing Editor has drafted this decision to help you prepare a revised submission.

Our expectation is that you will eventually carry out the additional experiments and report on how they affect the relevant conclusions either in a preprint on bioRxiv or medRxiv, or if appropriate, as a Research Advance in *eLife*, either of which would be linked to the original paper.

Summary:

We are excited by the novel technique that you developed which enables coincident visualization of ductular structures and their accompanying vasculature. The utility of the approach is well-demonstrated by your studies in a mouse model of Allagille's syndrome. The high quality of your data in that system support your claim that the technology could be extended to examine vasculogenesis and tubulogenesis in other organs and other diseases. This would represent an important technical advance and you have provided readers sufficient methodological detail to enable this.

Your current manuscript does have some limitations, however:

(1) Direct evidence for the generalizability of your success in the liver disease model is not presented in the current manuscript. Thus, including new data from at least one other organ/disease would strengthen your argument that the new technology is broadly applicable. The editors acknowledge that initiating new experiments to address this concern would significantly delay publication and so, are not insisting such studies be done prior to publication. However, the text should be edited to acknowledge the need for further research to confirm the utility of the new technology in other organs/diseases.

2) Based on your findings in adult mice, you draw conclusions about the pathogenesis of Allagille's syndrome that somewhat contradict current paradigms for that disease. Reviewer 2 has detailed new experiments that are necessary to validate your interpretation. Evaluation of additional human liver biopsy material would seem to be feasible in the short term. If the additional mouse studies cannot be accomplished in a sufficiently timely fashion, the current manuscript should be revised to acknowledge the limitations of the existing data set and need for further research to clarify disease pathogenesis.

As noted above:

1) Text should be edited to address both of the aforementioned concerns.

2) Liver histology and clinical data from additional Allagille's syndrome patients with less severe/earlier stage disease should be incorporated.

Revisions expected in follow-up work:

1) Additional experiments in the mouse model to evaluate ductal/vascular pathology at earlier stages of disease.

Reviewer #1:

This work is exciting because:

1) It provides a technological advance that is broadly applicable to studying effective and dysregulated tubulogenesis in multiple organs and preclinical models of various diseases.

2) The technology is novel because it permits simultaneous mapping of vascular and ductal morphogenesis – processes that are known to be linked but which have been challenging to examine coincidentally.

3) The new technique is clearly described and the authors provide appropriate caveats about is limitations, as well as its advantages, relative to current approaches that are being used to evaluate branching morphogenesis.

Reviewer #2:

Hankeova and colleagues describe a new technology named DUCT, which they have developed for 3D visualization and quantification of the architectural parameters of two tubular systems in a given organ simultaneously. The authors chose the biliary and portal vein trees in the mouse liver as their model for the development and assessment of DUCT. Comparison of DUCT with two other techniques used for the visualization of tubular structures (ink injection and iDISCO+) indicated a number of advantages for DUCT, although, as the authors acknowledge, using resins limits the diameter of smallest detectable tubes to 5 μm. They then used DUCT to investigate the patterns of biliary tree regeneration and the potential processes involved in the age-dependent improvement of the liver phenotypes in a mouse model of Alagille syndrome (ALGS). This model*, Jag1^Ndr/Ndr^* strain, was previously shown (by the authors) to have a rather severe cholestatic phenotype in the early postnatal period but a significant improvement of the phenotypes in adult animals, in agreement with other mouse models of this disease. The DUCT analysis showed a number of novel and somewhat unexpected features in the biliary tree of the adult *Jag1^Ndr/Ndr^* livers, suggesting potential mechanisms involved in the improvement of the liver phenotypes in some ALGS patients. In support of these observations, the authors observed similar features in the biliary system of several ALGS patients based on liver staining and 2D imaging.

Development of DUCT and the related data analysis pipeline reported in the current manuscript can be considered an important technological advance in the analysis of biliary development and regeneration and the relation between biliary and portal venous structures. The study has also provided potentially important insights into the mechanisms through which ALGS livers compensate for poor initial bile duct maturation. In addition, the DUCT technique has applications beyond liver development, namely the analysis of other organs with tubular structures, like vascular system, bronchi in the lung and urinary ducts in the kidney. Moreover, the authors provide a balanced Discussion by acknowledging that DUCT has a number of shortcomings that need to be addressed in future work (the above-mentioned 5 μm limit, the semi-automatic nature of the analysis, as opposed to a fully automated analysis, etc.).

A couple of major concerns need to be addressed:

1) Lack of analysis of an earlier time point in the *Jag1^Ndr/Ndr^* mouse:

The authors mention throughout the manuscript that their analysis of adult mouse livers show the outcome of the biliary system regeneration in these mice. However, they do not mention whether they have analyzed the livers of younger *Jag1^Ndr/Ndr^* mice before their phenotypic improvement. To use the parameters measured in adults as an indication for biliary tree regeneration, one will need to know how the biliary tree looked like at an earlier time point. Given the abnormal serum chemistry and liver histology in younger *Jag1^Ndr/Ndr^* mice (the authors' previous work), these animals will undoubtedly have biliary tree abnormalities at a young age. However, the specifics of those abnormalities are not known. Therefore, whether a given feature of the adult, "regenerated" tree is truly a compensatory phenomenon occurring during the improvement phase or whether it's part of the original biliary tree abnormality in these mice is not clear. The best and most conclusive way to address this issue would be to analyze 3-4 *Jag1^Ndr/Ndr^* animals at a young age, when there is still significant cholestasis. Such analysis will also determine whether there are intermediary structural steps that occur during regeneration. It will also determine how DUCT performs in livers with bile duct paucity.

It is possible that the resin casting would not work well in younger animals due to smaller liver size and biliary/portal system diameter. If because of this issue or due to COVID-related restrictions the authors are not able to address this point by performing the suggested experiment, they should clarify throughout the manuscript that these features are called regenerative in the manuscript based on the assumption that they do not exist in the younger *Jag1^Ndr/Ndr^* mice, and that without performing DUCT on livers from young animals, one cannot know for sure which one of the observed biliary tree features are responsible for the functional recovery in these animals.

2) Patient sample selection and analysis

Based on the description in the Materials and methods section, the patient samples seem to all come from severe cases (liver transplant). These are presumably patients whose biliary system was not able to (sufficiently) regenerate to avoid transplantation. Therefore, one can argue that they are not equivalent to the Ndr mice (which show phenotypic recovery). This needs to be highlighted in the Results section. To test whether the Ndr findings correspond to the recovery in human patients, one would ideally need to analyze ALGS liver biopsies from older (>5 year old) patients who did not need a transplant despite significant cholestasis early in life. However, those patients are not likely to undergo a liver biopsy once they show phenotypic recovery. Therefore, the next best thing might be to analyze liver biopsies from younger patients that based on their medical chart have gone on to show improvement later in life. Regardless of how the authors decide to address this issue, they should try to obtain and report additional information about the patients (like total and direct bilirubin) so that the readers can assess the severity of their cholestasis at the time of tissue harvest.*Reviewer #3:*

The authors describe an improvement of an existing resin casting technique which enables to visualize and quantify the 3D morphology of two distinct tubular (lumenized) structures in a single organ. The new method is applied to the characterization of portal veins and bile ducts in normal mouse livers and in livers affected with Alagille syndrome. The authors beautifully illustrate the morphology and spatial relationship between veins and ducts in normal and diseased livers. The quality of the 3D images is outstanding, the conclusions are well supported by the data, the text is very clearly written.

1) The authors' main goal is to describe a widely applicable method for investigating tubular structures. However, one may then expect that proof of principle would be provided for a second organ.

2) The work on normal and diseased livers is essentially descriptive and the authors provide good arguments supporting how the morphology of regenerated ducts, i.e. increased branching and tortuosity, could contribute to restoration of bile duct function in Alagille syndrome. It also unexpectedly uncovered that bile duct branching becomes less dependent on portal vein branching in diseased liver than in normal liver, or how bile duct tortuosity is increased in disease conditions. Unfortunately, the underlying mechanisms are not investigated.

---

## [Author Response]

[…] (1) Direct evidence for the generalizability of your success in the liver disease model is not presented in the current manuscript. Thus, including new data from at least one other organ/disease would strengthen your argument that the new technology is broadly applicable. The editors acknowledge that initiating new experiments to address this concern would significantly delay publication and so, are not insisting such studies be done prior to publication. However, the text should be edited to acknowledge the need for further research to confirm the utility of the new technology in other organs/diseases.

To strengthen our claim that DUCT can be utilized for other tubular organs, we now include data from a double resin casted lung. We visualised the respiratory system by injecting one resin into the trachea and the pulmonary arterial network by injecting into the pulmonary artery. The µCT segmented 3D reconstruction is presented in Figure 1—figure supplement 1E.

2) Based on your findings in adult mice, you draw conclusions about the pathogenesis of Allagille's syndrome that somewhat contradict current paradigms for that disease. Reviewer 2 has detailed new experiments that are necessary to validate your interpretation. Evaluation of additional human liver biopsy material would seem to be feasible in the short term. If the additional mouse studies cannot be accomplished in a sufficiently timely fashion, the current manuscript should be revised to acknowledge the limitations of the existing data set and need for further research to clarify disease pathogenesis.

We thank the reviewers for this comment and we agree with reviewer’s suggestion. To validate our interpretation, we analysed biopsies from 6 additional patients with mild Alagille syndrome that never went through liver transplantation, the overview of all patient samples is summarized in Figure 2—figure supplement 1. Importantly, one of these patients had been biopsied on three occasions from the neonatal period, via a regenerating period, to a post-regenerated stage with de novo generated bile ducts. We identified, in these liver biopsies from patients with mild Alagille syndrome, the same regenerative features as in *Jag1^Ndr/Ndr^* adult mice. These results are presented in new Figure 2C, Figure 3F, Figure 4E and Figure 6C.

Revisions expected in follow-up work:1) Additional experiments in the mouse model to evaluate ductal/vascular pathology at earlier stages of disease.

To address this comment, we applied DUCT to analyse P15 liver, a stage at which the *Jag1^Ndr/Ndr^* animals display mild jaundice and are cholestatic (new Figure 1C, Figure 1—figure supplement 5D). DUCT revealed severe bile duct paucity in most of the *Jag1^Ndr/Ndr^* lobes, but we detected a rudimentary biliary system in 6 out of 7 *Jag1^Ndr/Ndr^* livers imaged (Figure 1C and Figure 1—figure supplement 5B, C, E). Based on our previous data (Andersson et al., 2018) and current results we propose that the adult *Jag1^Ndr/Ndr^* biliary system is not established during embryogenesis and early postnatal stages, but forms postnatally between P-15 and adulthood, in a subset of animals. Due to the phenotype observed at P15 we decided to use the term “de novo generated” bile ducts for the *Jag1^Ndr/Ndr^* bile ducts throughout the manuscript.

Reviewer #2:[…] A couple of major concerns need to be addressed:1) Lack of analysis of an earlier time point in the Jag1^Ndr/Ndr^ mouse:The authors mention throughout the manuscript that their analysis of adult mouse livers show the outcome of the biliary system regeneration in these mice. However, they do not mention whether they have analyzed the livers of younger Jag1^Ndr/Ndr^ mice before their phenotypic improvement. To use the parameters measured in adults as an indication for biliary tree regeneration, one will need to know how the biliary tree looked like at an earlier time point. Given the abnormal serum chemistry and liver histology in younger Jag1^Ndr/Ndr^ mice (the authors' previous work), these animals will undoubtedly have biliary tree abnormalities at a young age. However, the specifics of those abnormalities are not known. Therefore, whether a given feature of the adult, "regenerated" tree is truly a compensatory phenomenon occurring during the improvement phase or whether it's part of the original biliary tree abnormality in these mice is not clear. The best and most conclusive way to address this issue would be to analyze 3-4 Jag1^Ndr/Ndr^ animals at a young age, when there is still significant cholestasis. Such analysis will also determine whether there are intermediary structural steps that occur during regeneration. It will also determine how DUCT performs in livers with bile duct paucity.It is possible that the resin casting would not work well in younger animals due to smaller liver size and biliary/portal system diameter. If because of this issue or due to COVID-related restrictions the authors are not able to address this point by performing the suggested experiment, they should clarify throughout the manuscript that these features are called regenerative in the manuscript based on the assumption that they do not exist in the younger Jag1^Ndr/Ndr^ mice, and that without performing DUCT on livers from young animals, one cannot know for sure which one of the observed biliary tree features are responsible for the functional recovery in these animals.

Thank you for this comment and experimental suggestion. Initially we had not performed DUCT on the younger animals due to major surgical challenges of performing the procedure in young mice, and due to the frailty of the *Jag1^Ndr/Ndr^* mice in both the vascular and biliary systems. Nevertheless, we agree that DUCT would be important to apply to younger mice, and therefore applied DUCT to 10 *Jag1^Ndr/Ndr^* animals at P15. The resin was possible to inject in 7 out of 10 *Jag1^Ndr/Ndr^* mice included in the experiment. The common bile duct was too narrow in the 3 uninjected *Jag1^Ndr/Ndr^* mice and could not accommodate the stretched PE10 tubing used for injections. The new data on P15 injected mice is presented in new Figure 1C, D, E and new Figure 1—figure supplement 5. DUCT showed a very rudimentary biliary system in P15 livers, and quantification of cleared resin-injected mice showed that while wild type mice have ca 75% coverage of the liver by a biliary system at this stage, *Jag1^Ndr/Ndr^* mice display 0-40% coverage. Interestingly, the coverage is inversely correlated with total bilirubin levels, supporting our later suggestion that 3D analysis of the biliary system may correlate better with bilirubin levels than 2D analysis.

2) Patient sample selection and analysisBased on the description in the Materials and methods section, the patient samples seem to all come from severe cases (liver transplant). These are presumably patients whose biliary system was not able to (sufficiently) regenerate to avoid transplantation. Therefore, one can argue that they are not equivalent to the Ndr mice (which show phenotypic recovery). This needs to be highlighted in the Results section. To test whether the Ndr findings correspond to the recovery in human patients, one would ideally need to analyze ALGS liver biopsies from older (>5 year old) patients who did not need a transplant despite significant cholestasis early in life. However, those patients are not likely to undergo a liver biopsy once they show phenotypic recovery. Therefore, the next best thing might be to analyze liver biopsies from younger patients that based on their medical chart have gone on to show improvement later in life. Regardless of how the authors decide to address this issue, they should try to obtain and report additional information about the patients (like total and direct bilirubin) so that the readers can assess the severity of their cholestasis at the time of tissue harvest.

We agree that it would be valuable to include patients with mild disease or recovered patients that did not go through liver transplantation. To address this question, we have analysed 8 liver biopsies from 6 patients with mild Alagille syndrome (see comment 2). Further, we have provided a table (Table 1) that shows the results from liver functional tests from all the patients included in the study and specified which sample type was used (biopsy vs. explant). We changed the labelling throughout the manuscript to distinguish patients that had liver recovery as M-ALGS (mild) and patients who went through liver transplantation are labelled as S-ALGS (severe). As mentioned above, one of the mild patients was biopsied as a neonate, during regeneration, and after regeneration. The mild/regenerated patients also show the phenotypes we identified with DUCT and in sections from *Jag1^Ndr/Ndr^* livers. These new data are presented in new Figure 2C, Figure 3F, Figure 4E and Figure 6C.

Reviewer #3:The authors describe an improvement of an existing resin casting technique which enables to visualize and quantify the 3D morphology of two distinct tubular (lumenized) structures in a single organ. The new method is applied to the characterization of portal veins and bile ducts in normal mouse livers and in livers affected with Alagille syndrome. The authors beautifully illustrate the morphology and spatial relationship between veins and ducts in normal and diseased livers. The quality of the 3D images is outstanding, the conclusions are well supported by the data, the text is very clearly written.

We thank the reviewer for the positive comments. We would like to note that the identification and use of two different resins, with different radiopacities, allows for semi-automated segmentation of the systems. Previous double injections, to our knowledge, used the same resin for both systems (therefore the previous existing resin casting technique was quite different).

1) The authors' main goal is to describe a widely applicable method for investigating tubular structures. However, one may then expect that proof of principle would be provided for a second organ.

Thank you for this comment, we agree and we now include resin casting of lung, new Figure 1—figure supplement 1E (please also see comment 1).

2) The work on normal and diseased livers is essentially descriptive and the authors provide good arguments supporting how the morphology of regenerated ducts, i.e. increased branching and tortuosity, could contribute to restoration of bile duct function in Alagille syndrome. It also unexpectedly uncovered that bile duct branching becomes less dependent on portal vein branching in diseased liver than in normal liver, or how bile duct tortuosity is increased in disease conditions. Unfortunately, the underlying mechanisms are not investigated.

We agree with the reviewer that it would be exciting to investigate the mechanisms underlying the abnormal bile duct behaviour in *Jag1^Ndr/Ndr^* mice and patents with Alagille syndrome. Investigating these mechanisms is however outside the scope of the current manuscript, which aimed to establish a new technique for 3D analysis, and provide new insights into the 3D architecture of wild type and *Jag1^Ndr/Ndr^* mice.